# DEBIASED DEEP EVIDENTIAL REGRESSION FOR VIDEO TEMPORAL GROUNDING

## ABSTRACT

Existing Video Temporal Grounding (VTG) models perform well in accuracy but often fail to address open-world challenges posed by open-vocabulary queries and out-of-distribution (OOD) videos, which can lead to unreliable predictions. To address uncertainty, particularly with OOD data, we build a VTG baseline using Deep Evidential Regression (DER), which excels in capturing both aleatoric and epistemic uncertainty. Despite promising results, our baseline faces two key biases in multimodal tasks: (1) Modality imbalance, where uncertainty estimation is more sensitive to the visual modality than the text modality; (2) Counterintuitive uncertainty, resulting from excessive evidence suppression in regularization and uneven sample error distribution in conventional DER. To address these, we propose an RFF block for progressive modality alignment and a query reconstruction task to enhance sensitivity to text queries. Additionally, we introduce a Geom-regularizer to debias and calibrate uncertainty estimation. This marks the first extension of DER in VTG tasks. Extensive experiments demonstrate the effectiveness and robustness of our approach. Our code will be released soon.

## 1 INTRODUCTION

Video is emerging as the primary information carrier in the streaming media era. With the influx of video data, the need for efficiently and precisely extracting desired video content is increasingly critical, leading to Video Temporal Grounding (VTG) emerging as a core research area in the field of computer vision Dosovitskiy et al. (2020); Li et al. (2023); Chen et al. (2024). While extensive research aimed at enhancing cross-modal reasoning to facilitate fine-grained and precise multi-modal alignment leads to significant advances in VTG Lin et al. (2023), few studies focus on the widespread uncertainties present in open-world scenarios Amini et al. (2020), which can be classified into epistemic uncertainty and aleatoric uncertainty. Epistemic uncertainty is mainly attributed to the Knowledge Gap as shown in Figure 2 (a). Specifically, user queries and video inputs often come from

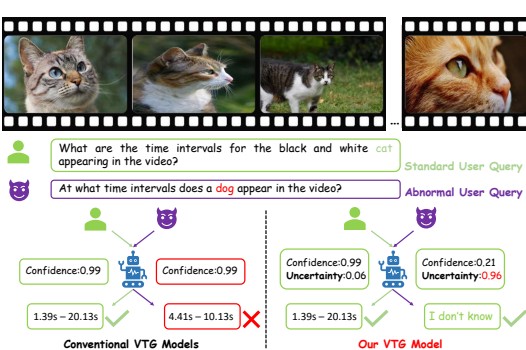

Figure 1: Conceptual illustration: Conventional models give random responses to OOD queries, unfit for critical decisions. In contrast, our model reliably delivers sensible, informed answers.

out-of-distribution (OOD) sources, diverging significantly from the in-distribution (ID) data used in training. This natural discrepancy creates a Knowledge Gap, making it challenging for models to accurately understand and respond to user needs. Moreover, semantic ambiguities can also hinder accurate contextual understanding and cause epistemic uncertainty, as shown in Figure 2 (b). Aleatoric uncertainty typically arises from inherent variability in training data, such as subjective annotation and variations in low-level features as shown in Figure 2 (c) and (d) respectively. Subjective annotations occur when different annotators provide varying queries and labels for the same sample, influenced by their personal views and habits. Variations in low-level features such as texture, edges, resolution,

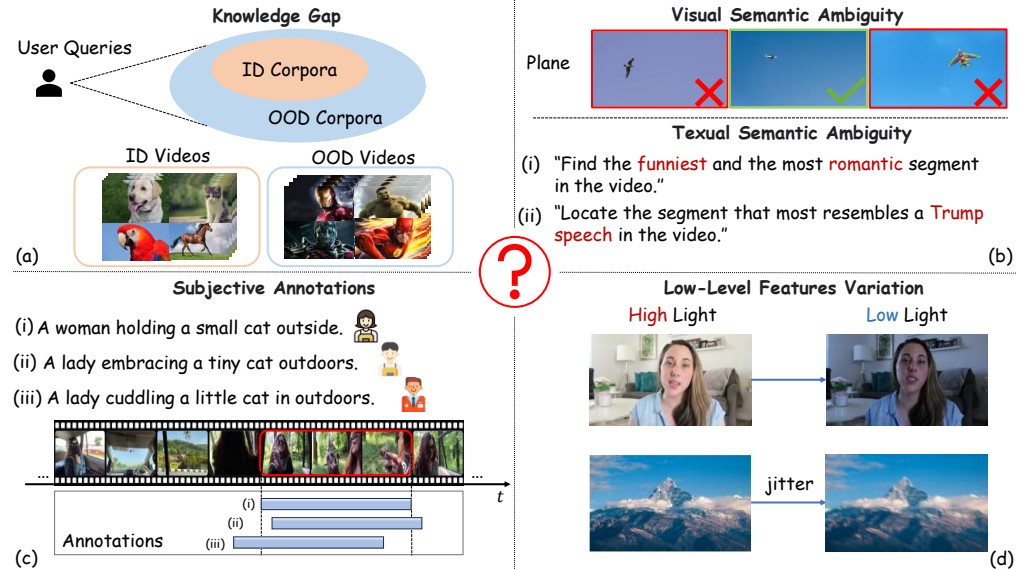

Figure 2: **Motivation illustration.** Epistemic uncertainty stems from knowledge gaps and semantic ambiguity in (a) and (b), while aleatoric uncertainty arises from subjective annotations and low-level feature variations in (c) and (d). In (a), the knowledge gap highlights the model's limitations due to insufficient real-world coverage, and (b) shows challenges from semantic ambiguity in visual and textual inputs. (c) emphasizes the subjectivity in annotations, while (d) illustrates uncertainty from lighting, resolution, jitter, and other low-level factors.

lighting, camera jittering, and scene transitions can contaminate the trainging video. Both of them can lead to aleatoric uncertainty in prediction for certain samples.

Unfortunately, when confronted with anomalies, such as the abnormal queries shown in Figure 1, conventional VTG models often respond with nearly random and indiscreet answers. These models fail to handle potential uncertainties in an open world appropriately. This inadequacy is detrimental in scenarios that require cautious decision-making, such as security and confidential environments. To address the limitations of VTG methods in open-world scenarios, we integrate Deep Evidential Regression (DER)Amini et al. (2020) into an uncertainty-aware baseline. Based on Evidential Deep Learning (EDL)Sensoy et al. (2018), DER excels at capturing uncertainty, particularly with OOD data. This makes DER a natural fit for VTG tasks, where open-vocabulary queries and OOD videos present significant challenges.

However, due to the complexity of the VTG task and limitations of the vanilla DER method, the baseline exposes two critical biases: (1) Modality imbalance: Uncertainty estimation disproportionately favors the visual modality over text. This stems from the nature of VTG, where visual input dominates tasks like keyframe identification. DER's loss function, optimized for continuous visual signals, further emphasizes this imbalance. Moreover, simple modality concatenation or unimodal self-attention Lei et al. (2021a); Lin et al. (2023) fails to foster sufficient interaction between text and visual features, leading to over-reliance on visual input. (2) Counterintuitive uncertainty: Higher-error predictions sometimes receive lower uncertainty due to DER's regularizer limitations. Unlike classification-based EDL methods, DER lacks a standard KL-divergence term, relying instead on a heuristic regularizer that overly suppresses evidence, especially in low-error samples. This misaligns uncertainty estimates, with low-error samples showing higher uncertainty and vice versa.

To address modality imbalance, we propose a Debiased DER Model for Video Temporal Grounding (**DDM-VTG**). Specifically, it corporates with a Reflective Flipped Fusion (RFF) block with dual branches for progressive cross-modal alignment, along with a query reconstruction (QR) task to strengthen the text branch. This enhances the model's sensitivity to text and mitigates bias in uncertainty estimation. To resolve counterintuitive uncertainty, we introduce a simple yet effective Geom-regularizer that adjusts uncertainty estimation based on prediction accuracy, adaptively suppressing overconfidence and debiasing the system. Our contributions are three-fold: (**1**) This is the first extension of DER to the VTG task, establishing an uncertainty-aware baseline. Leveraging DER's uncertainty estimation, we effectively address open-world challenges in VTG. (**2**) We

identified and addressed two critical biases in the baseline—modality imbalance and counterintuitive uncertainty—through the RFF block, an auxiliary QR task, and the Geom-regularizer, which together calibrate uncertainty estimation. (**3**) Extensive experiments show our method's effectiveness, robustness, and interpretability across multiple benchmarks.

## 2 RELATED WORK

### 2.1 VIDEO TEMPORAL GROUNDING

Video Temporal Grounding (VTG) identifies correlated shots in videos based on natural language queries, which broadly supports various downstream video comprehension tasks, such as video moment retrieval Anne Hendricks et al. (2017); Chen et al. (2018); Zhang et al. (2020); Moon et al. (2023b); Li et al. (2024), highlight detection Rui et al. (2000); Sun et al. (2014); Moon et al. (2023b), and video summarization Gygli et al. (2014); Jiang & Mu (2022); Mahasseni et al. (2017); Nalla et al. (2020); Sharghi et al. (2017); Wu et al. (2022). These tasks generally involve formulating the boundaries of significant semantic segments Jiang & Mu (2022); Moon et al. (2023b); Lin et al. (2023). Numerous innovative and effective methods are developed to address the challenges in VTG. For instance, CTRL Anne Hendricks et al. (2017) and MCN Gao et al. (2017) initially generate proposals using sliding windows, which they rank in terms of a cross-modal matching score. MomentDETR Lei et al. (2021b) applies a transformer to predict potential moments through learnable queries. Furthermore, QD-DETR Moon et al. (2023b) employs a cross-attention module and a negative pair training scheme to enhance multi-modal alignment. MomentDiff Li et al. (2024) initially sets random boundaries for temporal segments and iteratively refines them to better match the intended semantics. However, existing approaches typically yield deterministic predictions, operating under the assumption that semantic segments are demarcated by clear and precise boundaries. This presumption neglects the inherent ambiguity and uncertainty associated with determining the true extents of these segments. To address this gap, we explicitly models and quantify the semantic uncertainty of video segment boundaries.

### 2.2 UNCERTAINTY LEARNING

Recent studies have highlighted inherent ambiguities and biases in VTG datasets, which significantly impact the integrity and performance of models Zhou et al. (2021); Zhang et al. (2023). These uncertainties are categorized into annotation and query uncertainties. Annotation uncertainty stems from varying temporal boundaries assigned by different annotators to the same query, while query uncertainty arises from the use of differing descriptions for the same video moment, underscoring the subjective nature of video interpretation Zhou et al. (2021); Zhang et al. (2023). Furthermore, these datasets exhibit pronounced biases, with common events being overly represented and a small subset of queries accounting for most actions, leading to a skewed distribution that creates a long tail in ground-truth timestamps Zhang et al. (2023); Otani et al. (2020). These findings underscore the need for meticulous curation of datasets and the adoption of uncertainty-aware modeling approaches Arnab et al. (2020); Malinin & Gales (2018); Zhou & Levine (2021); Gawlikowski et al. (2023). Among the techniques for modeling uncertainty, Evidential Deep Learning (EDL) has shown promise. Originating from the principles of Dempster-Shafer Theory Shafer (1992) and Subjective Logic Sensoy et al. (2018); Jøsang (2016), EDL models uncertainty explicitly through the distribution of "second-order probabilities" over network outputs, finding applications across various classification tasks including action recognition Bao et al. (2021), multi-view clustering Han et al. (2020; 2022), and semantic segmentation Holmquist et al. (2023) *etc*. Leveraging DER Amini et al. (2020) for its extension, EDL has effectively been applied to regression tasks such as stereo matching Wang et al. (2022) and emotion attributes estimation Wu et al. (2023). Nevertheless, DER faces challenges like evidence contraction due to the non-negativity of prior parameters in the Normal Inverse-Gamma (`NIG`) distribution. New regularizers have been developed to address these issues, enhancing reliability and performance Wu et al. (2024). However, DER often encounters gradient disappearance in high uncertainty areas, necessitating ongoing refinement of its regularization methods Ye et al. (2024); Meinert et al. (2023). In this study, We introduce DER into the VTG task to manage uncertainties in open-world inputs, while further addressing the modality imbalance it presents and improving the structural flaws of the vanilla DER regularizer. To the best of our knowledge, this marks the first successful attempt to extend DER in VTG task.

## 3 PRELIMINARIES

DER Amini et al. (2020) places evidential priors over the original Gaussian likelihood function and trains the model to infer the hyperparameters of the evidential distribution. This approach enables the model to learn both aleatoric and epistemic uncertainty. In our context, adjacent video frames often exhibit similar semantics, which introduces uncertainty in precisely locating temporal boundaries. The start or end temporal boundary of a video is represented by distinct Gaussian distributions: $\boldsymbol{b} \sim \mathcal{N}(\mu, \sigma^2)$, where $\boldsymbol{b} \in \mathbb{R}^{1 \times \mathcal{H}}$ represents the start or end of moments observed $\mathcal{H}$ times. We assume that observations of the same type (either all starts or all ends) are *i.i.d.*. The corresponding expectation $\mu$ and variance $\sigma^2$ of the Gaussian distribution subject to `NIG` prior:

$$p(\mu, \sigma^2 \mid \underbrace{\gamma, \upsilon, \alpha, \beta}_{\boldsymbol{\varphi}}) = \mathcal{N}(\mu | \gamma, \sigma^2 \upsilon^{-1}) \Gamma^{-1}(\sigma^2 | \alpha, \beta), \tag{1}$$

where $\boldsymbol{\varphi} = (\gamma, \upsilon, \alpha, \beta)$ are the prior `NIG` distribution parameters derived from the video content and user queries, serve as conditionals for the Gaussian estimates of $b_i$ with $\gamma \in \mathbb{R}, \upsilon > 0, \alpha > 1, \beta > 0$. The gamma function is denoted by $\Gamma(\cdot)$. We use a linear evidential predictor to estimate $\boldsymbol{\varphi}$, training it to maximize the likelihood. Since the likelihood function has a form of Student-t distribution ($\mathrm{St}$), we minimize the negative logarithmic likelihood (NLL) as follows:

$$\mathcal{L}_i^{\mathrm{NLL}} = -\log p(b_i | \boldsymbol{\varphi}) = -\log \left( \mathrm{St}\left( b_i; \gamma, \frac{\beta(1 + \upsilon)}{\upsilon \alpha}, 2\alpha \right) \right), \tag{2}$$

Models optimized only on observed samples with the NLL loss (*i.e.* Eq. 2) tend to overfit and exhibit overconfidence. To counter this, DER introduced a regularizer for the $i$-th prediction as follows:

$$\mathcal{L}_i^{\mathrm{R}}(\boldsymbol{\vartheta}) = \Delta \cdot \Phi, \tag{3}$$

where $\Delta = |b_i - \gamma|$ represents the error, $\Phi = 2\upsilon + \alpha$ denotes the evidence, and $\boldsymbol{\vartheta}$ are the model parameters, with $b_i$ as the ground truth. Detailed formulation can be found in Appendix A. Using the `NIG` distribution, prediction, aleatoric and epistemic uncertainties are calculated as follows:

$$\underbrace{\mathbb{E}[\mu] = \gamma}_{\text{prediction}}, \quad \underbrace{\mathbb{E}[\sigma^2] = \frac{\beta}{\alpha - 1}}_{\text{aleatoric}}, \quad \underbrace{\mathrm{Var}[\mu] = \frac{\beta}{\upsilon(\alpha - 1)}}_{\text{epistemic}}, \tag{4}$$

$\mathbb{E}[\sigma^2]$ refers to the inherent noise in the data, which cannot be reduced or eliminated. $\mathrm{Var}[\mu]$ reflects the model's lack of confidence in its own predictions due to limited knowledge.

## 4 METHODOLOGY

### 4.1 PROBLEM DEFINITION

Given a video $V = \{v_i\}_{i=1}^{L_v}$ and a query $Q = \{q_i\}_{i=1}^{L_q}$, each represented as vectors in $\mathbb{R}^D$ where $L_v$ and $L_q$ denote the counts of clips and tokens respectively, the task of VTG is to assign each clip a label $b$ based on its relevance to $Q$. $b$ can be of multiple types, i.e., a time span $m_i = [m_i^s, m_i^e]$ for **Moment Retrieval**, a saliency score $s_i \in [0, 1]$ for **Highlight Detection**, or a binary value $f_i \in \{0, 1\}$ for **Video Summarization**. The choice of label type depends on the specific VTG subtask being addressed, allowing for flexible application across various video understanding scenarios.

### 4.2 BUILDING BASELINE WITH VANILLA DER FOR VTG

As illustrated in Figure 3 (a), we first build an uncertainty-aware baseline by integrating DER into the VTG task. The motivation for this is to address the inherent challenges of open-world scenarios, such as handling OOD data and open-vocabulary queries. Overall, the loss function of the model can be formulated as follows:

$$\mathcal{L}_i^{\mathrm{B}}(\boldsymbol{w}) = \lambda_{\mathrm{NLL}} \mathcal{L}_i^{\mathrm{NLL}} + \lambda_{\mathrm{Reg}} \mathcal{L}_i^{\mathrm{R}}(\boldsymbol{w}), \tag{5}$$

$$\mathcal{L}_{base} = \mathcal{L}_G + \lambda_{\mathrm{der}} \frac{1}{N} \sum_{i=1}^{N} \mathcal{L}_i^{\mathrm{B}}(\boldsymbol{w}), \tag{6}$$

where $N$ symbolizes the number of clips in a training set and $\mathcal{L}_G$ denotes VTG loss. While DER effectively estimates uncertainty, its vanilla form presents limitations like modality imbalance and flawed uncertainty estimation, which our baseline exposes and sets the foundation for improvement.

Figure 3: Comparison of the baseline (a) and our improved model (b) for the VTG task. In (a), the baseline model demonstrates a lack of sensitivity to textual information due to the overlap between the VTG task's nature and the DER objective function design, leading to a heavy reliance on visual features. Furthermore, structural flaws in the vanilla DER's regularizor contribute to unreliable uncertainty estimates. In (b), the RFF block and Query Reconstruction (QR) head enhance text sensitivity through deeper cross-modality interaction, while the Geom-regularizer addresses vanilla DER's structural flaws and achieves more trustworthy uncertainty estimation.

## 4.3 DEBIASED DER MODEL FOR VTG

To address the biased uncertainty estimation in the baseline in section 4.2, caused by modality imbalance and counterintuitive uncertainty, we propose DDM-VTG. Our model introduces a RFF block for progressive cross-modal alignment, reducing over-reliance on visual features, and a QR task to enhance text sensitivity. As shown in Figure 3 (b), DDM-VTG first encodes an untrimmed video and masked query, reconstructs the masked tokens via the RFF block, and performs VTG. The debiased DER head assesses both aleatoric and epistemic uncertainties, while the VTG and QR heads manage task stages. Further details are provided in subsequent sections.

**RFF block.** The RFF block processes inputs from the video and text branches, alternating the roles of video and text as queries and keys/values using shared parameters. Through the cross-attention module, initial features $V^{(1)}$ and $Q^{(1)}$ of video and text branches are respectively updated, reflecting each other's information. Following cross-attention, each branch refines its features through self-attention to enhance internal feature representation. The outputs of self-attention serve as inputs to the next iteration of the RFF block, progressively enhancing modal alignment. This process is applied sequentially from block 1 to block $n$ in terms of Eq. (7). After completing the $n$-th layer, the refined video and query features are output. The specific workflow process is detailed in Appendix E.

$$V^{(i+1)} = SA_v^{(i)}(CA_{q\to v}^{(i)}), \quad Q^{(i+1)} = SA_q^{(i)}(CA_{v\to q}^{(i)}), \quad i = 1, 2, \ldots, n-1 \tag{7}$$

**VTG head**. The VTG head features three distinct modules for tasks outlined in section 4.1. For Video Summarization, the output from the frozen video encoder undergoes three 1x3 Conv layers, each with a ReLU activation. The Moment Retrieval head is similar but outputs two channels for offsets. Highlight Detection uses attentive pooling to form a sentence representation from query tokens, then computes the saliency score between video tokens and query as their cosine similarity. Details for each module and corresponding loss are available in the Appendix F.

**Query reconstruction task**. To ensure robust cross-modal alignment capabilities, during the initial phase of alignment, entities within the query are masked at a specified ratio. This approach compels the model to leverage contextual information available from the corresponding video and the remaining unmasked tokens in the query. Through the QR head, the model infers and reconstructs the masked tokens. The loss function associated with this process, aimed at optimizing the model's

cross-modal inference capabilities, is outlined below:

$$\mathcal{L}_{qr} = \mathbb{E}\left[-\sum_{i=1}^{l}\log P(w_i \mid U, V)\right], \tag{8}$$

where $l$ represents the number of masked tokens, $w_i$ the $i$-th masked token, $U$ the unmasked tokens providing linguistic context, and $V$ the video features that enhance cross-modal contextual understanding for accurate token prediction. After the warm-up in the first phase, in the next phase, the QR head is frozen and $\mathcal{L}_{qr}$ is not computed.

**Geom-regularization**. The heuristic regularizor (*i.e.* Eq.(3)) in conventional DER aims to mitigate overconfidence by suppressing evidence, particularly for samples with high error. However, excessive suppression can lead to underconfidence due to non-adaptive suppression and sample imbalance. To be clear, we first consider the minus gradient of $\mathcal{L}_i^{\mathrm{R}}$ for $\Phi$ as follows:

$$-\nabla_\Phi \mathcal{L}_i^{\mathrm{R}} = -\Delta, \tag{9}$$

To explore the penalties bias in the vanilla regularizer, we visualized its optimization direction by examining the gradient field derived from the Eq.(9). As shown in Fig 4 (a), the gradient is solely linked to the error and not to the evidence, indicating that the model cannot ascertain when the evidence has been adequately suppressed. This approach often results in insufficient gradients for batches dominated by small errors, potentially leading to biased penalties on evidence. As the model converges, the dominance of low error samples with small gradients skews the batch's average gradient. Consequently, their evidence is over-suppressed, while high error samples see their evidence neglected or ad versely adjusted, as shown in Appendix C.4.

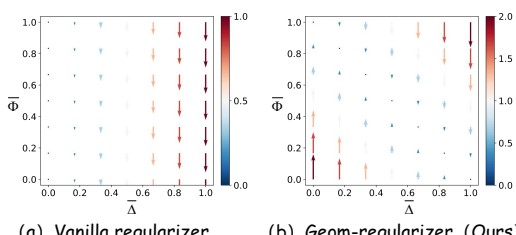

(a) Vanilla regularizer    (b) Geom-regularizer (Ours)

Figure 4: Gradient field comparison. (a) Vanilla regularizer applies penalties based solely on error, with a tendency to decrease evidence as error increases. (b) Our Geom-regularizer modulates penalties dynamically based on both error magnitude and current evidence levels. Our approach reflects the principle that accurate predictions should have higher evidence, while evidence should be suppressed for less accurate predictions.

To overcome these limitations, we introduce Geom-regularization, inspired by Amini et al. (2020), promoting the principle that "***accurate predictions should have high evidence, while inaccurate ones should have low evidence***". This approach provides more rational constraints rather than merely suppressing evidence. Initially, we normalize $\Delta$ to $\overline{\Delta}$ and $\Phi$ to $\overline{\Phi}$ (*i.e.* Appendix B.2), which ensures that the model assigns $\overline{\Phi} = 1$ to samples with $\overline{\Delta} = 0$, and $\overline{\Phi} = 0$ to samples with $\overline{\Delta} = 1$. We then ensure that the points $(\overline{\Delta}, \overline{\Phi})$ closely follow the line $\overline{\Phi} + \overline{\Delta} = 1$ using a line regularizer as below:

$$\mathcal{L}_i^{\mathrm{L}}(\boldsymbol{w}) = \|\overline{\Phi} + \overline{\Delta} - 1\|_2^2, \tag{10}$$

we can follow the analysis for $\mathcal{L}_i^{\mathrm{R}}$. The minus gradient of $\mathcal{L}_i^{\mathrm{L}}$ with respect to $\overline{\Phi}$ as below:

$$-\nabla_{\overline{\Phi}} \mathcal{L}_i^{\mathrm{L}} = -2(\overline{\Delta} + \overline{\Phi} - 1), \tag{11}$$

which indicates this simple regularizer offers a gradient that relates to both error and evidence, enabling adaptive evidence suppression, as illustrated in Figure 4 (b).

Our training objective for the evidential head is the combination of NLL and Geom-regularization:

$$\mathcal{L}_i^{\mathrm{e}}(\boldsymbol{w}) = \lambda_{\mathrm{NLL}}\mathcal{L}_i^{\mathrm{NLL}} + \lambda_{\mathrm{geom}}\mathcal{L}_i^{\mathrm{L}}(\boldsymbol{w}), \tag{12}$$

To this end, our total loss can be formulated by a combination of common grounding loss $\mathcal{L}_G$ (discussed in Appendix F ) and our evidential loss:

$$\mathcal{L} = \mathcal{L}_G + \lambda_{\mathrm{der}}\frac{2}{N}\sum_{i=1}^{N}\mathcal{L}_i^{\mathrm{e}}(\boldsymbol{w}) + \mathcal{L}_{qr}, \tag{13}$$

where $N$ symbolizes the number of clips in a training set.

Table 1: Performance on QVHighlights, TACoS, and Charades-STA datasets. **Bold** numbers indicate the best performance, and underlined numbers indicate the second best performance. MR denotes Moment Retrieval, and HD denotes Highlight Detection.

| Method | QVH-MR | | | QVH-HD | | TACoS | | | Charades-STA | | |
|---|---|---|---|---|---|---|---|---|---|---|---|
| | R@0.5 | R@0.7 | Avg.M | MAP | HIT@1 | R@0.5 | R@0.7 | mIoU | R@0.5 | R@0.7 | mIoU |
| M-DETR Lei et al. (2021a) | 53.9 | 34.8 | 32.2 | 35.7 | 55.6 | 28.0 | 12.9 | 27.2 | 46.0 | 27.5 | 41.3 |
| UMT Liu et al. (2022a) | 60.3 | 44.3 | 38.6 | 39.9 | **64.2** | 23.5 | 13.2 | 25.0 | 42.7 | 24.1 | 41.6 |
| QD-DETR Moon et al. (2023a) | 62.7 | 46.7 | 41.2 | 39.1 | 63.0 | 24.7 | 12.0 | 25.5 | 52.1 | 30.6 | 45.5 |
| UniVTG Lin et al. (2023) | 59.7 | - | 36.1 | 38.8 | 61.8 | 35.0 | 17.4 | 33.6 | 58.0 | 35.7 | 50.1 |
| EaTR Jang et al. (2023) | 61.4 | 45.8 | 41.7 | 37.2 | 58.7 | - | - | - | - | - | - |
| MomentDiff Li et al. (2024) | 57.4 | 39.7 | 36.0 | - | - | 33.7 | - | - | 55.6 | 32.4 | - |
| **DDM-VTG (Ours)** | **65.0** | **49.4** | **43.0** | **40.1** | 63.4 | **37.3** | **19.4** | **33.9** | **60.2** | **38.0** | **51.6** |

## 5 EXPERIMENT

We focus on the following key considerations to conduct convincing experiments: **1)** Despite focusing on robustness and interpretability, does our proposed DDM-VTG model demonstrate competitive performance relative to current state-of-the-art VTG models? **2)** Does the proposed QR task and RFF blocks enhance performance in VTG tasks? **3)** Does DDM-VTG give low uncertainty when performing high localization accuracy statistically, and vice versa? **4)** Is our proposed Geom-regularizer more robust than the vanilla regularizer (*i.e.*, Eq. (3))? **5)** Can the model output a high uncertainty score in various OOD scenarios to inform abnormality?

### 5.1 DATASETS AND IMPLEMENTATION DETAILS

**Datasets**. We conducted experiments on several widely used public datasets from diverse scenes: Charades-STA Gao et al. (2017) (in-door activities), QVHighlights Lei et al. (2021b) (untrimmed daily vlogs & news), TACoS Regneri et al. (2013) (cooking scenes), and TVSum (YouTube videos). The detailed information including specific task domains and sizes for different datasets is reported in Appendix B.1 Table 4 with their different hyperparameters.

**Metrics**. For moment retrieval, we use recall@1 with IoU thresholds of 0.5 and 0.7, mean average precision (MAP) with IoU thresholds of 0.5 and 0.75, and MAP avg, which is the average MAP across IoU thresholds from 0.5 to 0.95 in 0.05 increments. For Highlight detection and Video summarization, we use MAP. Following Lei et al. (2021a), an additional metric HIT@1 is utilized for the Highlight detection task in the QVHighlights dataset, representing the hit ratio of the highest-scored clip.

**Experimental Settings**. Following previous works Lin et al. (2023); Li et al. (2024), we utilize CLIP Radford et al. (2021) (ViT-B/32) and SlowFast Feichtenhofer et al. (2019) (ResNet-50) as a frozen backbone. Unless otherwise specified, the number (**n**) of RFF blocks is set to 4. The training process is divided into two stages. In the first stage, QR masks and reconstructs noun entities in the query. Each sentence has 1 noun masked by default. The default epoch for QR is set to 30, with a learning rate of 1e-5. We utilize spaCy's transformer-based parser- Honnibal & Montani (2017) to extract noun entities from the query text, and the masking is accomplished by zeroing out the noun entities at the embedding level. In the second stage, DDM-VTG predicts bounding boxed on the visual branch at each video clip. Since our purpose of using DER is to optimize uncertainty without affecting the model's grounding capability, the gradient of delta in Eq. (10) is set to zero. To avoid similar predictions, we utilize NMS with a threshold of 0.7 at evaluation to achieve better performance. If not stated otherwise, we used the line regularizer on the evidential head. All training for the moment retrieval tasks are conducted on four Tesla V100 GPUs. For the video summarization task, due to the smaller scale of the TVSum dataset, we used only a single V100 GPU.

### 5.2 QUANTITATIVE RESULTS

**Comparison with the state-of-the-art.** To demonstrate the fundamental capabilities of DDM-VTG, we compared it with several state-of-the-art methods across multiple benchmarks. As reported in Table 1, DDM-VTG outperforms existing methods on various metrics. On the QVHighlights val split, for the Moment Retrieval task, DDM-VTG obtains 65.0% for R1@0.5 and 49.4% for R1@0.7. For the Highlight Detection task, DDM-VTG reaches 63.4% for HIT@1. In the Moment Retrieval task, DDM-VTG outperforms MomentDiff Li et al. (2024) by 8.0% on R1@0.5, and in

Table 2: Video summarization results on TVsum. † denotes methods with audio modality.

| Method | VT | VU | GA | MS | PK | PR | FM | BK | BT | DS | Avg. |
|---|---|---|---|---|---|---|---|---|---|---|---|
| LIM-S Xiong et al. (2019) | 55.9 | 42.9 | 61.2 | 54.0 | 60.3 | 47.5 | 43.2 | 66.3 | 69.1 | 62.6 | 56.3 |
| Trailer Wang et al. (2020) | 61.3 | 54.6 | 65.7 | 60.8 | 59.1 | 70.1 | 58.2 | 64.7 | 65.6 | 68.1 | 62.8 |
| SL-Module Xu et al. (2021) | 86.5 | 68.7 | 74.9 | **86.2** | 79.0 | 63.2 | 58.9 | 72.6 | 78.9 | 64.0 | 73.3 |
| MINI-Net Hong et al. (2020)† | 80.6 | 68.3 | 78.2 | 81.8 | 78.1 | 65.8 | 57.8 | 75.0 | 80.2 | 65.5 | 73.2 |
| UMT Liu et al. (2022b)† | **87.5** | 81.5 | 88.2 | 78.8 | 81.4 | **87.0** | **76.0** | 86.9 | 84.4 | **79.6** | 83.1 |
| UniVTG Lin et al. (2023) | 83.9 | 85.1 | **89.0** | 80.1 | 84.6 | 81.4 | 70.9 | 91.7 | 73.5 | 69.3 | 81.0 |
| **DDM-VTG (Ours)** | 85.4 | **93.0** | 92.5 | 81.4 | **87.2** | 79.6 | 72.0 | 92.2 | 87.1 | 75.3 | **84.6** |

Table 3: Ablation studies on the QVHighlights validation split. (a) $\text{Var}_{\text{vis}}$ and $\text{Var}_{\text{text}}$ represent uncertainty variance across noise levels, with noise gradually added to the respective modality inputs. The variance indicates the sensitivity of the DDM-VTG model to different modal inputs. (b) compares the effectiveness of model estimation uncertainty using different regularizers. **Bold** denotes the best performance, and underline indicates the second best.

(a) Component ablation.

| Method | R@0.5 | $\text{Var}_{\text{vis}}$ | $\text{Var}_{\text{text}}$ | $\Delta_{\text{Var}} \downarrow$ |
|---|---|---|---|---|
| Baseline | 61.1 | 9.17 | 0.85 | 8.32 |
| + RFF block | 62.4 | 8.63 | 1.60 | 7.03 |
| + QR | 63.8 | 4.89 | 3.91 | 0.98 |
| Full Model | **65.0** | 4.85 | 5.54 | **0.69** |

(b) Uncertainty effectiveness metrics

| Method | EUCM $\downarrow$ | Entropy $\uparrow$ |
|---|---|---|
| Baseline | 0.32 | 0.09 |
| Vanilla-Reg. | 0.31 | **0.35** |
| Geom-Reg. | **0.29** | 0.31 |

the Highlight Detection task, it surpasses EaTR Jang et al. (2023) by 4.7% on HIT@1. We also conduct performance comparison on the TACoS and Charades-STA datasets. On TACoS, DDM-VTG outperforms the nearest competitor UniVTG by 2.3% and 2.0% on R1@0.5 and R1@0.7, respectively. For Charades-STA, DDM-VTG surpasses UniVTG by 2.2%, 2.3%, and 1.5% across three metrics.

We also validate DDM-VTG's performance in video summarization. We present a comparison on the TVSum dataset in Table 2. For each domain in TVSum, DDM-VTG has generally demonstrated strong performance. Specifically, in the VU domain, DDM-VTG outperforms UniVTG by 8%. Additionally, on the overall average metrics, DDM-VTG exceeds UniVTG by 3.6% and surpasses the UMT model, which utilizes audio modality, by 1.5%.

**Ablation study.** To demonstrate the effectiveness of our proposed debiasing method, we design targeted metrics to quantify the degree of debiasing in the model, and conduct extensive ablation on the validation set of QVHighlights Lei et al. (2021b). For modality imbalance, we define a $\text{Var}_{\text{vis}}$, $\text{Var}_{\text{text}}$ and $\Delta_{\text{Var}}$ to measure the sensitivity of uncertainty under varying levels of noise interference. To be detailed, we add different level of Gaussian noise to video embeddings, and replace a varying proportion of text tokens with irrelevant text, where the noise schedule is the same as in Figure 6. We evaluate the variance of the uncertainty for a video query pair under different noise levels and then take the average across all samples. We use values of $\text{Var}_{\text{vis}}$ and $\text{Var}_{\text{text}}$ calculated in this way to quantify the uncertainty sensitivity for vision and text modality respectively, and the difference between them $\Delta_{Var}$ to describe the modality balance under different model settings. As reported in Table 3 (a) , our results strongly demonstrate the effectiveness of both RFF block and QR task.

For counterintuitive uncertainty, we define a new metric Error-Uncertainty Consistency Measure (EUCM), with more details can be found in Appendix D. Moreover, we also compute the information entropy of different uncertainty distributions, which is used to evaluate the expressive ability of the evidential predictor, as reported in Table 3 (b) . The entropy of predictions with the vanilla regularizer is greater than that with the Geom-regularizer. However, as demonstrated in section 5.3 predictions made with the vanilla regularizer are prone to be misleading. Consequently, even though it exhibits higher information entropy, this entropy may encompass substantial "biased information". In contrast, the Geom-regularizer not only achieves higher information entropy but also results in a lower EUCM score, indicating its superior performance.

**Parameter analysis.** As shown in Figure 5(a), we examine the change in MAP as $\lambda_{\text{geom}}$ and $\lambda_{\text{der}}$ gradually increased. We observe that the MAP reaches its optimal value when $\lambda_{\text{geom}}$ is set to $10^{-2}$, which we therefore set $\lambda_{\text{geom}}$ to $10^{-2}$. Also, when $\lambda_{\text{der}}$ is small, the model's performance remains unaffected. However, as $\lambda_{\text{der}}$ increases to $1 \times 10^{-2}$, MAP begins to decline, reaching its lowest at $1 \times 10^{-1}$. This can be explained by the fact that an excessively high uncertainty constraint weight

forces the model's optimization direction to overfit the evidential head rather than maintaining its basic grounding ability. Therefore, we set $\lambda_{der}$ to $1 \times 10^{-3}$. Figure 5 (b) shows the model performance differences under different settings of query reconstruction task's epochs and learning rate. Notably, when the number of QR epochs increased from 0 to 50, there was a significant improvement in MAP (+1.5%). Besides, we observe the best QR learning rate (lr) of $1 \times 10^{-4}$. Appendix B provided more details on parameters setting across multiple benchmarks.

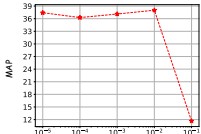 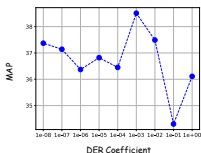 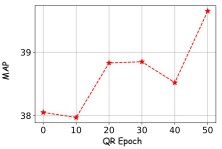 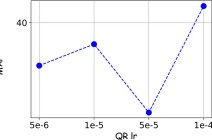

(a) Performance on different $\lambda_{geom}$ and $\lambda_{der}$.  (b) Performance on different QR epochs and lr.

Figure 5: Parameters Analysis on QVHighlights val split. We examined the change of MAP. (a) Evaluate the effectiveness of our proposed Geom-regularizer (left) and der loss (right) under different weights. (b) Demonstrates the impact of the query reconstruction task at different epochs (left) and learning rates (right).

### 5.3 QUALITIVE RESLUTS

**Uncertainty sensitivity to modalities.** To more clearly demonstrate the role of the proposed RFF block and QR task in promoting modality balance, we conduct fair adversarial experiments on the QVHighlights validation set to compare the modality sensitivity of the baseline model and DDM-VTG. We apply the noise addition method mentioned in ablation and used the noise intensity format shown in Figure 6. Also as illustrated in Figure 6, when progressively increasing noise levels for two different modalities, the output uncertainty of the baseline model (Top row) shows high sensitivity to the visual modality and low sensitivity to the textual modality. In

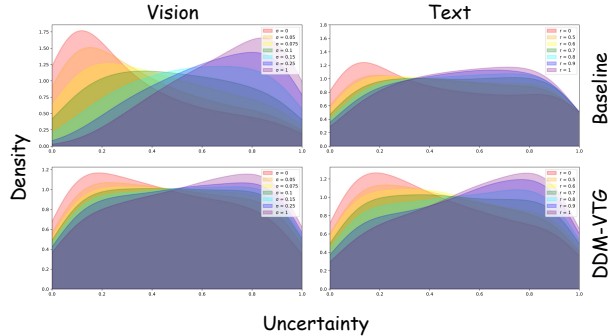

Figure 6: **Uncertainty KDE over differect noise level**. Using Gaussian kernel density estimation (KDE), We plotted the uncertainty distribution for the QVHighlighte val set.

contrast, the DDM-VTG model (Bottom row) achieves balanced uncertainty across both modalities. Regardless of which modality the noise is added to, the uncertainty distribution of the model shifts from left-skewed to right-skewed at the same rate.

**Uncertainty calibration.** Since we propose geom-regularizor to calibrate of counterintuitive uncertainty prediction, we aim to assess the efficacy of our approach by contrasting the performance of aleatoric and epistemic uncertainty estimation with and without our regularization technique, as well as against using vanilla regularization in Amini et al. (2020). Ideally, optimal uncertainty measures should effectively identify deviations in predictions (*i.e.*, take high uncertainty when the model is making errors). Figure 7 illustrates our comparison of different regularization methods on the QVHighlights validation set. The horizontal axis of each scatter plot represents $\overline{\Delta}$ (*i.e.* normalized error), while the vertical axis represents one of the two types of uncertainty. More discussion is provided in the Appendix C.2 and C.4.

**Temporal bias sensitivity.** As previously reported in studies, most Moment Retrieval datasets exhibit significant imbalances in the duration and position of moments. As shown in Figure 8 (a), using the QVHighlights dataset as an example, we visualize the joint distribution of the normalized start times and end times of all ground truth moments. The light-colored areas in the figure indicate regions with almost no moment distribution, leading to **Temporal OOD**. Higher epistemic uncertainty is demanded when samples belong to the Temporal OOD region. We analyzed the uncertainty predicted by the model in different time regions under various experimental settings in the QVHighlights

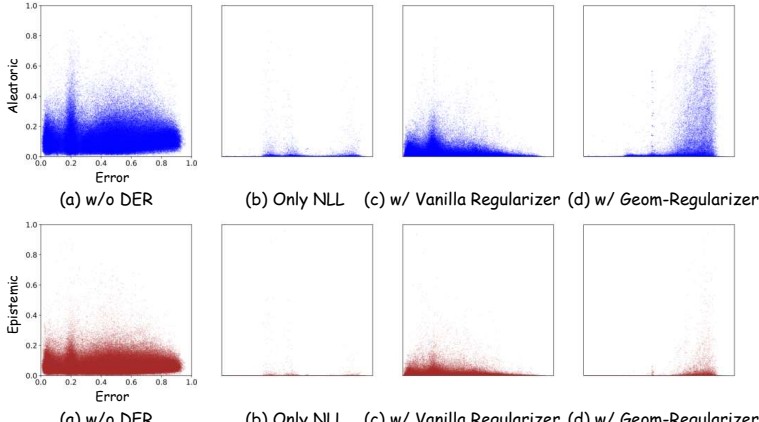

Figure 7: Effects of Various Regularization Techniques on Uncertainty Distribution. (a)-(d) illustrate the impact of different regularization methods on the relationship between aleatoric uncertainty (top row) and epistemic uncertainty (bottom row) with respect to prediction error. The models include: (a) without DER, (b) only NLL, (c) with Vanilla Regularizer, and (d) with Geom-Regularizer.

dataset. Figure 8 (b) shows that without DER constraints, the evidential head's predicted aleatoric and epistemic uncertainty tends to simply fit the biased temporal distribution. Figure 8 (c) shows that using only the NLL constraint (*i.e.*, Eq. 2), most regions exhibit extremely low epistemic uncertainty, indicating the model is overly confident in its predictions. Figure 8 (d) illustrates that with Vanilla regularization, although this approach does suppress the concentration of uncertainty in a specific region, it does not show sensitivity to OOD data. Figure 8 (e) demonstrates that with our proposed Geom-regularizor, the temporal OOD regions exhibit significantly higher epistemic uncertainty.

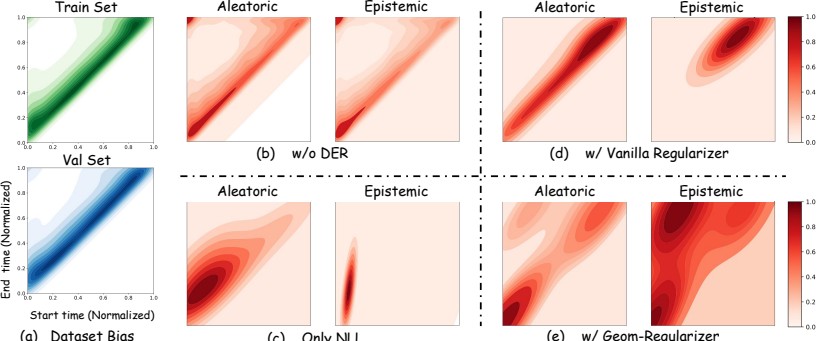

Figure 8: **Dataset bias sensitivity.** (a) Joint distributions of the start and end timestamps of the ground-truth moments in the QVHighlights dataset. (b), (c), (d), and (e) show the predicted uncertainty's sensitivity to temporal biases in the dataset under different conditions.

**Cases study.** We have meticulously selected several examples of model inference under the typical scenarios described in Figure 2 to further validate the effectiveness of the DDM-VTG model. The relevant details are provided in Appendix C.3 and C.5.

# 6 CONCLUSION

As the development of Artificial General Intelligence (AGI) progresses, increasingly sophisticated VTG models are emerging. However, these models often falter when confronted with open-ended user inputs. Addressing this challenge, this paper introduces a robust VTG model, namely DDM-VTG, that not only possesses VTG capabilities but also overcomes two types of bias present in proposed baseline methods and enables useful and meaningful quantification of potential uncertainties. This allows the model to provide credible responses to queries that exceed its operational scope. Limited by data quality and scale, the model's capabilities are not particularly notable. Nevertheless, it offers strategies for enhancing the trustworthiness of AI decisions. Future research will focus on further expanding the decision-making reliability and interpretability of multimodal large language models in video-related downstream tasks.

## REPRODUCIBILITY STATEMENT

In adherence to ICLR's principles, the authors have made significant efforts to ensure the reproducibility of our research findings. The primary focus has been on providing a clear and detailed description of our methodology, which includes the novel Debiased Deep Evidential Regression for Video Temporal Grounding (DDM-VTG) model, the Reflective Flipped Fusion (RFF) block, and the Geom-regularizer. These are elaborated in Sections 4 and 5 of the main text. Furthermore, we have meticulously documented our data processing steps and experimental setup in Appendix B, which details the parameters of the datasets used and the configurations for model training. Also, we are committed to open-sourcing our code upon publication. This will allow fellow researchers to directly access the implementation details and reproduce the results as presented in our paper. We believe that these measures will substantially contribute to the reproducibility of our work and encourage further scientific inquiry in this domain.

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

## A  DERIVATIONS

### A.1  NORMAL INVERSE-GAMMA MOMENTS

We assume our data was drawn from a Gaussian with unknown mean and variance, $(\mu, \sigma^2)$. We probabilistically model these parameters, $\boldsymbol{\theta}$, according to:

$$\mu \sim \mathcal{N}(\gamma, \sigma^2 v^{-1}) \tag{14}$$

$$\sigma^2 \sim \Gamma^{-1}(\alpha, \beta). \tag{15}$$

Therefore, the prior joint distribution can be written as:

$$p(\underbrace{\mu, \sigma^2}_{\theta} \,|\, \underbrace{\gamma, v, \alpha, \beta}_{\varphi}) = p(\mu)\,p(\sigma^2) \tag{16}$$

$$= \mathcal{N}(\gamma, \sigma^2 v^{-1})\,\Gamma^{-1}(\alpha, \beta) \tag{17}$$

$$= \frac{\beta^\alpha \sqrt{v}}{\Gamma(\alpha)\sqrt{2\pi\sigma^2}} \left(\frac{1}{\sigma^2}\right)^{\alpha+1} \exp\left\{-\frac{2\beta + v(\gamma - \mu)^2}{2\sigma^2}\right\}. \tag{18}$$

The first-order moments of this distribution represent the maximum likelihood prediction as well as uncertainty (both aleatoric and epistemic).

$$\mathbb{E}[\mu] = \int_{\mu=-\infty}^{\infty} \mu\,p(\mu)\,\mathrm{d}\mu = \gamma \tag{19}$$

$$\mathbb{E}[\sigma^2] = \int_{\sigma^2=0}^{\infty} \sigma^2\,p(\sigma^2)\,\mathrm{d}\sigma^2 \tag{20}$$

$$= \int_{\sigma=0}^{\infty} \sigma^2\,p(\sigma^2)\,(2\sigma)\,\mathrm{d}\sigma \tag{21}$$

$$= \frac{\beta}{\alpha - 1}, \qquad \forall\,\alpha > 1 \tag{22}$$

$$\mathrm{Var}[\mu] = \int_{\mu=-\infty}^{\infty} \mu^2\,p(\mu)\,\mathrm{d}\mu - (\mathbb{E}[\mu])^2 \tag{23}$$

$$= \gamma^2 - \frac{\sigma^2}{v} - (\mathbb{E}[\mu])^2 \tag{24}$$

$$= \gamma^2 - \frac{\frac{\beta}{\alpha-1}}{v} - \gamma^2 \tag{25}$$

$$= \frac{\beta}{v(\alpha - 1)}, \qquad \forall\,\alpha > 1 \tag{26}$$

In summary,

$$\underbrace{\mathbb{E}[\mu] = \gamma}_{\text{prediction}}, \qquad \underbrace{\mathbb{E}[\sigma^2] = \frac{\beta}{\alpha-1}}_{\text{aleatoric}}, \qquad \underbrace{\mathrm{Var}[\mu] = \frac{\beta}{v(\alpha-1)}}_{\text{epistemic}}. \tag{27}$$

### A.2  MODEL EVIDENCE & TYPE II MAXIMUM LIKELIHOOD LOSS

In this subsection, we derive the posterior predictive or model evidence (*i.e.* Eq. 28) of a `NIG` distribution. Marginalizing out $\mu$ and $\sigma$ gives our desired result:

$$p(b_i|\boldsymbol{\varphi}) = \mathrm{St}\left(b_i; \gamma, \frac{\beta(1 + v)}{v\,\alpha}, 2\alpha\right). \tag{28}$$

$$p(b_i|\boldsymbol{\varphi}) = \int_{\boldsymbol{\theta}} p(b_i|\boldsymbol{\theta})p(\boldsymbol{\theta}|\boldsymbol{\varphi}) \, \mathrm{d}\boldsymbol{\theta} \tag{29}$$

$$= \int_{\sigma^2=0}^{\infty} \int_{\mu=-\infty}^{\infty} p(b_i|\mu,\sigma^2)p(\mu,\sigma^2|\boldsymbol{\varphi}) \, \mathrm{d}\mu \, \mathrm{d}\sigma^2 \tag{30}$$

$$= \int_{\sigma^2=0}^{\infty} \int_{\mu=-\infty}^{\infty} p(b_i|\mu,\sigma^2)p(\mu,\sigma^2|\gamma,v,\alpha,\beta) \, \mathrm{d}\mu \, \mathrm{d}\sigma^2 \tag{31}$$

$$= \int_{\sigma^2=0}^{\infty} \int_{\mu=-\infty}^{\infty} \left[ \sqrt{\frac{1}{2\pi\sigma^2}} \exp\left\{ -\frac{(b_i-\mu)^2}{2\sigma^2} \right\} \right] \tag{32}$$

$$\left[ \frac{\beta^\alpha\sqrt{v}}{\Gamma(\alpha)\sqrt{2\pi\sigma^2}} \left(\frac{1}{\sigma^2}\right)^{\alpha+1} \exp\left\{ -\frac{2\beta+v(\gamma-\mu)^2}{2\sigma^2} \right\} \right] \mathrm{d}\mu \, \mathrm{d}\sigma^2 \tag{33}$$

$$= \int_{\sigma^2=0}^{\infty} \frac{\beta^\alpha\sigma^{-3-2\alpha}}{\sqrt{2\pi}\sqrt{1+1/v}\Gamma(\alpha)} \exp\left\{ -\frac{2\beta+\frac{v(b_i-\gamma)^2}{1+v}}{2\sigma^2} \right\} \mathrm{d}\sigma^2 \tag{34}$$

$$= \int_{\sigma=0}^{\infty} \frac{\beta^\alpha\sigma^{-3-2\alpha}}{\sqrt{2\pi}\sqrt{1+1/v}\Gamma(\alpha)} \exp\left\{ -\frac{2\beta+\frac{v(b_i-\gamma)^2}{1+v}}{2\sigma^2} \right\} 2\sigma \, \mathrm{d}\sigma \tag{35}$$

$$= \frac{\Gamma(1/2+\alpha)}{\Gamma(\alpha)} \sqrt{\frac{v}{\pi}} \left(2\beta(1+v)\right)^\alpha \left(v(b_i-\gamma)^2+2\beta(1+v)\right)^{-(\frac{1}{2}+\alpha)} \tag{36}$$

$$p(b_i|\boldsymbol{\varphi}) = \mathrm{St}\left(b_i;\gamma,\frac{\beta(1+v)}{v\,\alpha},2\alpha\right). \tag{37}$$

$\mathrm{St}\left(\boldsymbol{b};\mu_{\mathrm{St}},\sigma_{\mathrm{St}}^2,v_{St}\right)$ is the Student-t distribution evaluated at $\boldsymbol{b}$ with location parameter $\mu_{\mathrm{St}}$, scale parameter $\sigma_{\mathrm{St}}^2$, and $v_{\mathrm{St}}$ degrees of freedom. Using this result we can compute the negative log-likelihood loss, $\mathcal{L}_i^{\mathrm{NLL}}$, for sample $i$ as:

$$\mathcal{L}_i^{\mathrm{NLL}} = -\log p(b_i|\boldsymbol{\varphi}) \tag{38}$$

$$= -\log\left( \mathrm{St}\left(b_i;\gamma,\frac{\beta(1+v)}{v\,\alpha},2\alpha\right) \right) \tag{39}$$

$$\mathcal{L}_i^{\mathrm{NLL}} = \tfrac{1}{2}\log\left(\tfrac{\pi}{v}\right) - \alpha\log(\Omega) + \left(\alpha+\tfrac{1}{2}\right)\log((b_i-\gamma)^2 v+\Omega) + \log\left(\tfrac{\Gamma(\alpha)}{\Gamma(\alpha+\frac{1}{2})}\right) \tag{40}$$

where $\Omega = 2\beta(1+v)$.

## B  DATASETS AND IMPLEMENTATION DETAILS.

### B.1  PARAMETERS OF DATASETS

In Table 4, we list the datasets used in this study, including dataset size, task category, video clip length, and detailed hyperparameters used for model training.

Table 4: VTG dtasets list. **MR** denotes Moment Retrieval, **HD** denotes Highlight Detection, and **VS** denotes Video Summarization. **S** means seconds, **LR** denotes learning rate, **Epo** denotes total training epochs, **Warm-up** means number of warm-up iterations, and **LR Drop** means the epoch that drops learning rate by $1/10$.

| Dataset | MR | HD | VS | # Samples | S | BS | LR | Epo | Warm-up | LR Drop | QR LR | QR Epo |
|---|---|---|---|---|---|---|---|---|---|---|---|---|
| QVHighlights | ✓ | ✓ | | 10.3K | 2 | 32 | $1e^{-4}$ | 200 | 10 | 180 | $1e^{-4}$ | 30 |
| Charades-STA | ✓ | | | 16.1K | 1 | 32 | $1e^{-4}$ | 100 | 10 | 50 | $1e^{-5}$ | 10 |
| TACoS | ✓ | | | 18.2K | 2 | 32 | $1e^{-4}$ | 150 | 10 | 80 | $1e^{-5}$ | 30 |
| TVSum | | | ✓ | 50 | 2 | 4 | $1e^{-3}$ | 400 | 50 | N/A | $1e^{-5}$ | 10 |

### B.2 IMPLEMENTATIONS FOR NORMALIZATIONS

**Normalization:** We have tried two normalization operations, *i.e.* min-max normalization and using activation function to normalize.

- **Min-Max normalization:** Assume we have evaluated an increasing sequence of errors, that is:

$$\{\Delta_1, \Delta_2, \cdots, \Delta_n\} \tag{41}$$

  where $n$ represents batch size. Min-Max Normalization maps $\Delta_i$ to $\overline{\Delta_i}$ by:

$$\overline{\Delta_i} = \frac{\Delta_i}{\Delta_n - \Delta_1} \tag{42}$$

  We recommend using this normalization method in training and batch testing.

- **Normalization using activation functions:** Use activation functions $tanh(\cdot)$ so that we can map $\Delta_i$ to $\overline{\Delta_i}$, which is between 0 and 1:

$$\overline{\Delta_i} = tanh(\Delta_i) \tag{43}$$

  And $\Phi$ is normalized in the same way to $\overline{\Phi}$. We recommend this normalization for single-point or small-batch testing.

**Histogram equalization:** Although we normalize the uncertainty, we still find that the distribution of uncertainty is extremely biased to 0. We consider that this is still due to the overconfidence effect that NLL brings to the model. In order to obtain a more expressive uncertainty estimation in the inference process, sometimes we use histogram equalization to post-process the normalized uncertainty.

---

**Algorithm 1** Histogram Equalization

---

**Input:** Sequence of values: $X = \{\Delta_1, \Delta_2, \ldots, \Delta_n\}$
**Output:** Equalized sequence: $X' = \{\overline{\Delta_1}, \overline{\Delta_2}, \ldots, \overline{\Delta_n}\}$
 1: $hist \leftarrow$ calculate_histogram$(X)$
 2: $cdf \leftarrow$ calculate_cdf$(hist)$
 3: $X' \leftarrow$ apply_cdf_mapping$(X, cdf)$
 4: **return** $X'$

---

## C QUALITATIVE ANALYSIS

### C.1 VISULIZATION OF ATTENTION MAP

The visualizations provided in Figure 9 and Figure 10 demonstrate the effectiveness of our DDM-VTG in achieving fine-grained cross-modal alignment in VTG. Figure 9 effectively maps the attention to the visual cues of a woman speaking, aligning precisely with the textual description, thereby enhancing video-to-text translation accuracy. Similarly, Figure 10 shows that DDM-VTG is capable of focusing on a group of friends interacting around a table, accurately reflecting the descriptive text, which is essential for generating contextually accurate video summaries. These examples underscore the potential of DDM-VTG not only to improve downstream task performance by ensuring temporal and contextual relevance but also to serve as a basis for investigating the model's uncertainty. By analyzing where the model allocates attention, researchers can identify areas of high confidence and potential uncertainty, aiding in the refinement of VTG models for more reliable and transparent AI-driven applications.

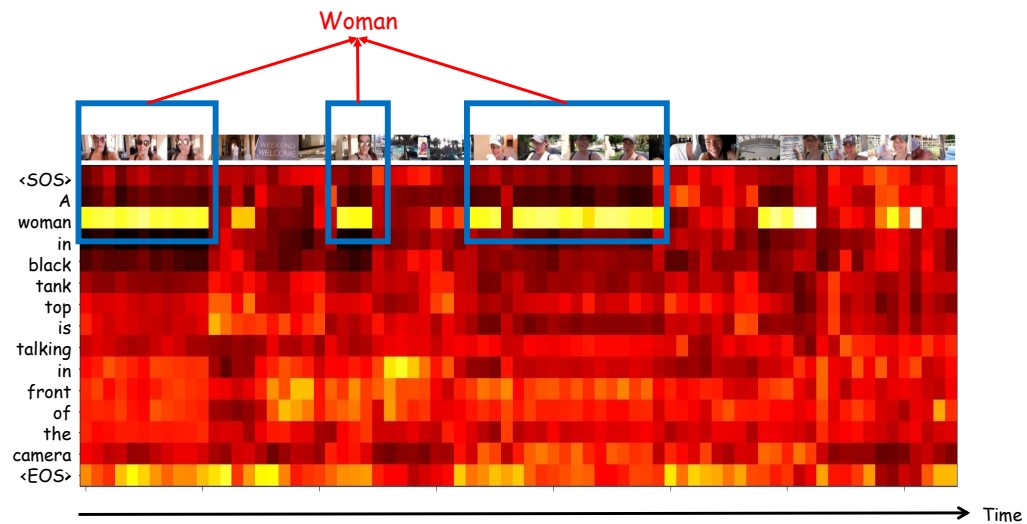

Figure 9: **Case I of attention map visualization.**

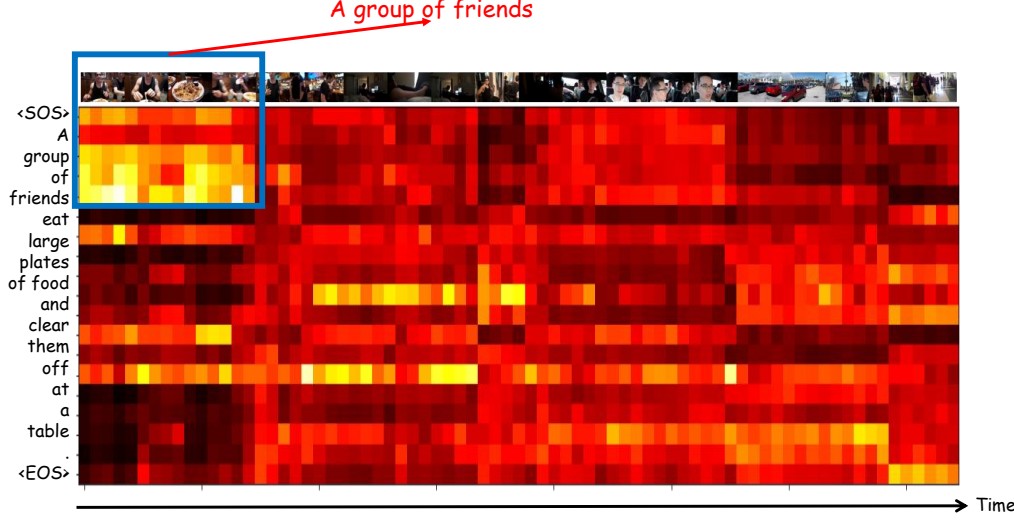

Figure 10: **Case II of attention map visualization.**

## C.2 VISUALIZATION OF UNCERTAINTY CALIBRATION

Figure 7 demonstrates the influence of different regularization strategies on model uncertainty in relation to prediction error. The top row focuses on aleatoric uncertainty, which is inherent data uncertainty, whereas the bottom row examines epistemic uncertainty, which stems from model ignorance. And we can discern the following key information:

- **(a) Without DER**: This model lacks any form of uncertainty management in the absence of DER, leading to inference results that are difficult to trust due to the complete absence of handling latent uncertainties.

- **(b) Only NLL**: In this configuration, the model exhibits extremely low uncertainty across all levels of error rates, indicating overconfidence due to overfitting. This overconfidence suggests a model that is not realistically cautious about its predictions.

- **(c) With Vanilla Regularizer**: Although the vanilla regularizer in DER measures and manages uncertainty, it paradoxically induces the model to express higher uncertainty at lower error rates and very low uncertainty at higher error rates. This counterintuitive

behavior is clearly problematic, as it does not align with rational expectations of uncertainty behavior.

- **(d) With Geom-Regularizer**: Compared to (a), our proposed Geom-regularizer effectively measures and manages uncertainty, enabling the model to indicate higher uncertainty at higher error rates and vice versa. Relative to (b), it successfully mitigates the model's overconfidence, which is beneficial for making prudent decisions. Against (c), it accurately calibrates the measurement of uncertainty, achieving a more sensible and intuitive assessment of uncertainty levels.

### C.3 CASES STUDY

In figure 11, we select some cases from the validation set of QVHighlights that support the effectiveness of our model. For example, we can easily observe that highly accurate predictions are often accompanied with very low uncertainty, while highly inaccurate predictions are accompanied with very high uncertainty, as shown by the first case and the last two cases. Additionally, when there exist scene changes (case 2, case 3) or changes in lighting conditions (case 5) in the video, the model is also prone to output higher uncertainty, especially aleatoric uncertainty.

### C.4 VISULIZATION OF ERROR-EVIDENCE EVOLUTION

As illustrated in Figure 12, it is obvious that "***accurate predictions with high evidence while inaccurate predictions with low evidence***" has been reflected in the knowledge of model with only NLL. Unfortunately, the vanilla regularizer excessively suppress the evidence of low error predictions, but ignores and even enlarges the evidence of high error predictions. Geom-regularizer turn the situation around, retain the main knowledge learned by NLL, and provides calibration for more reasonable uncertainty estimation.

### C.5 ADVERSARIAL EXPERIMENTS

We conduct adversarial experiments on DDM-VTG at the statistical level and case level, in order to demonstrate that DDM-VTG really capture increased predictive uncertainty on samples that have been adversarily perturbed.

For the case level, we design four experiments to show DDM-VTG is able to capture a high degree of uncertainty in some specific situations that is very likely to exist in reality, which has been discussed in Figure 2. In Figure 13, DDM-VTG demonstrate effective perception of ambiguous visual semantics, providing predictions while also outputting higher uncertainty. In Figure 14, DDM-VTG assign higher uncertainty to the OOD video (which is a cartoon) , even though both videos contain the semantic "a wolf is running". In Figure 15, DDM-VTG also assign higher uncertainty to the OOD video, which is infrared thermal imaging video. In Figure 16, the word "funny" in query is abstract and confuses the model, but DDM-VTG successfully provides high uncertainty to compensated for the failure in prediction.

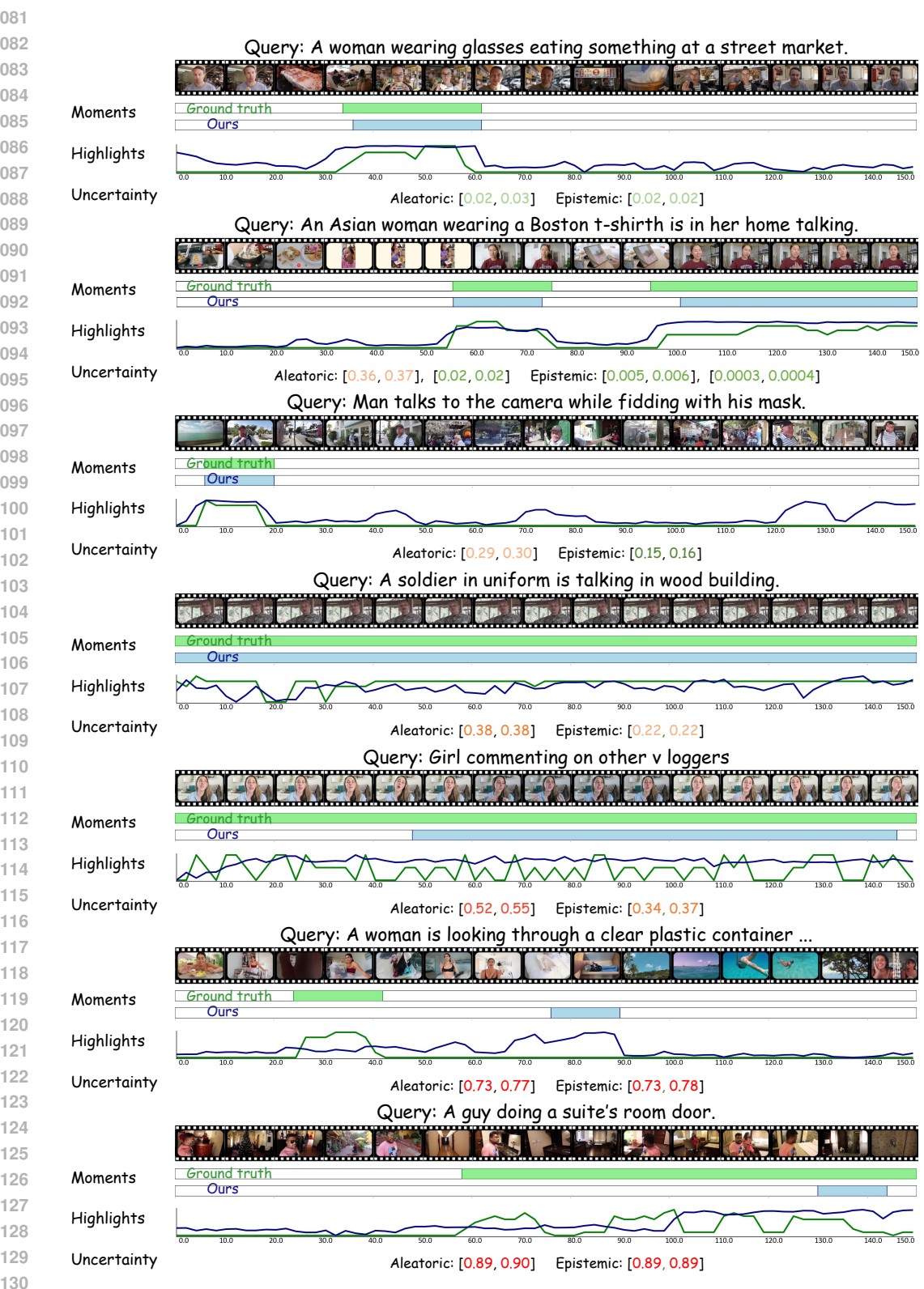

Figure 11: **Cases Study**

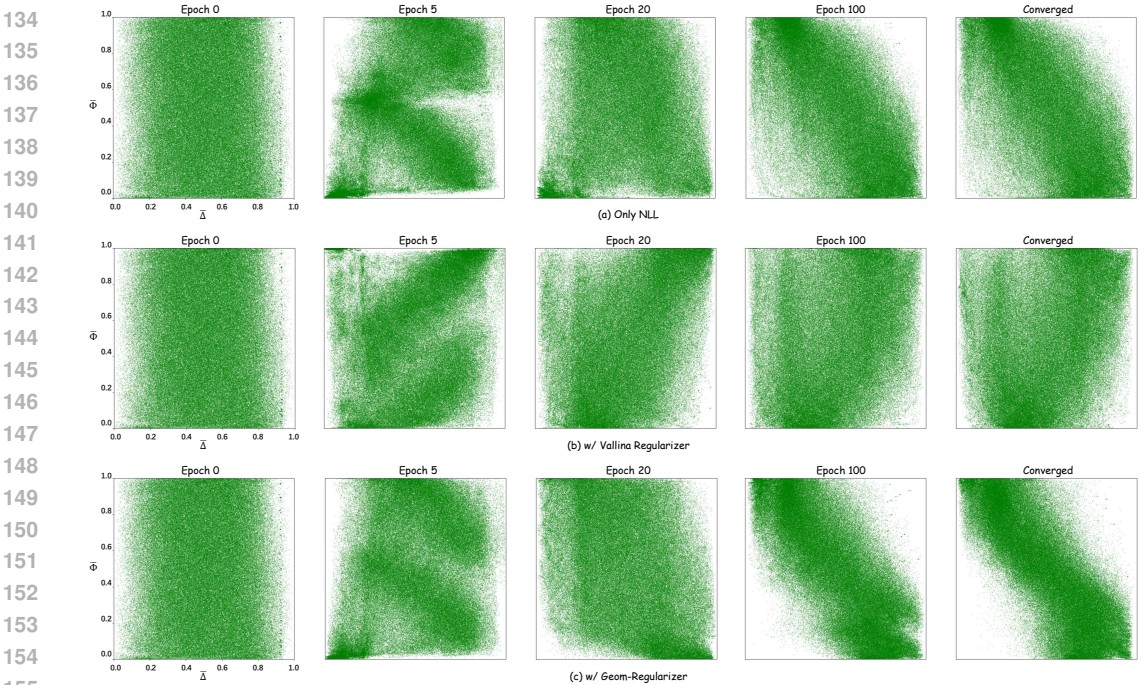

Figure 12: **Evolution of the predicted $(\overline{\Delta}, \overline{\Phi})$s' distribution over training epochs with different regularization techniques on QVHighlights Lei et al. (2021b).** This figure showcases how the evidence $(\overline{\Phi})$ and error $(\overline{\Delta})$ distributions evolve across training epochs (0, 5, 20, 100, and convergence) under three regularization strategies: (a) only NLL, (b) added vanilla regularizer, and (c) our Geom-regularizer.

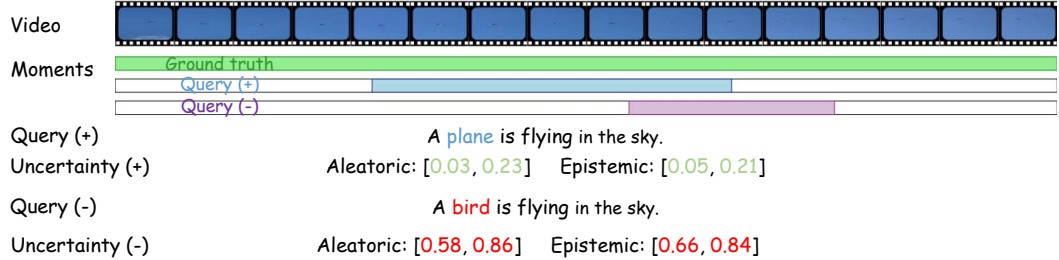

Query (+)      A plane is flying in the sky.
Uncertainty (+)      Aleatoric: [0.03, 0.23]    Epistemic: [0.05, 0.21]
Query (-)      A bird is flying in the sky.
Uncertainty (-)      Aleatoric: [0.58, 0.86]    Epistemic: [0.66, 0.84]

Figure 13: **Adversarial Case I.** We select a semantically ambiguous video, where the plane is extremely small, making it difficult for even humans to discern whether it is an airplane or a bird. We provide both the correct query (query with "plane") and an incorrect query (replacing "plane" with "bird") to DDM-VTG, compare the uncertainty of their highest confidence predictions, and find that the model assign higher uncertainty to the incorrect query.

## D QUANTITATIVE ANALYSIS OF GEOM-REGULARIZER

To evaluate how well the predicted uncertainty under different regularization settings aligns with the principle that **"larger errors should correspond to greater uncertainty"**, we introduce the Error-Uncertainty Consistency Measure (EUCM). EUCM is calculated as:

$$EUCM = \|\overline{\Delta} + \overline{\mathcal{U}}\|_2^2, \tag{44}$$

where $\mathcal{U}$ represents uncertainty. Moreover, we also compute the information entropy of different uncertainty distributions, which is used to evaluate the expressive ability of the evidential predictor.

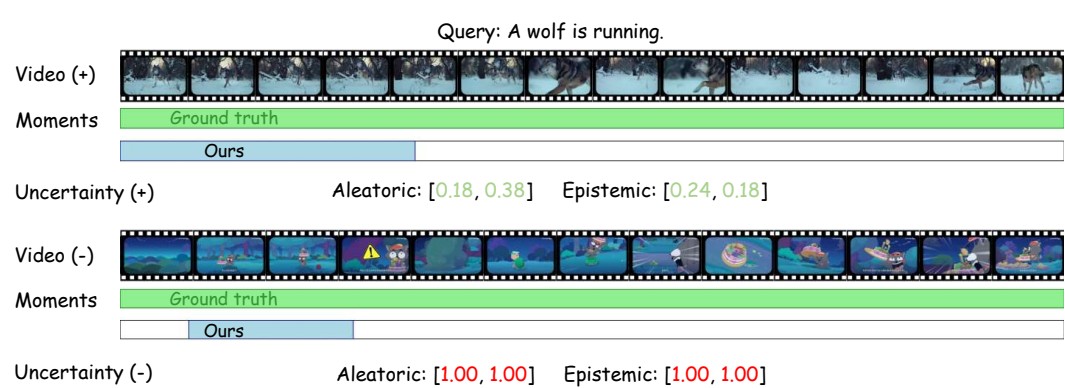

Figure 14: **Adversarial Case II.** We select a real video and an animated video of "a running wolf" and provide the model with the same query "a wolf is running". It can be observed that DDM-VTG outputs higher certainty for the animated video.

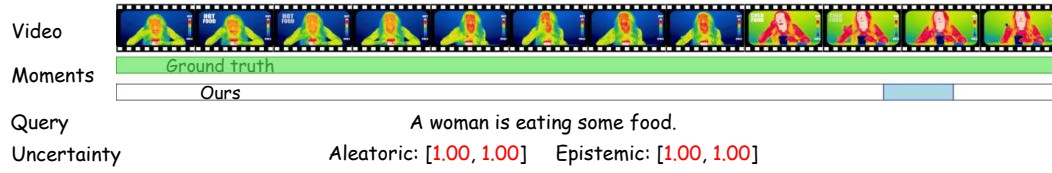

Figure 15: **Adversarial Case III.** We select an infrared thermal imaging video, a type that the model has rarely encountered in the training set. As an OOD video, DDM-VTG assign it very high uncertainty.

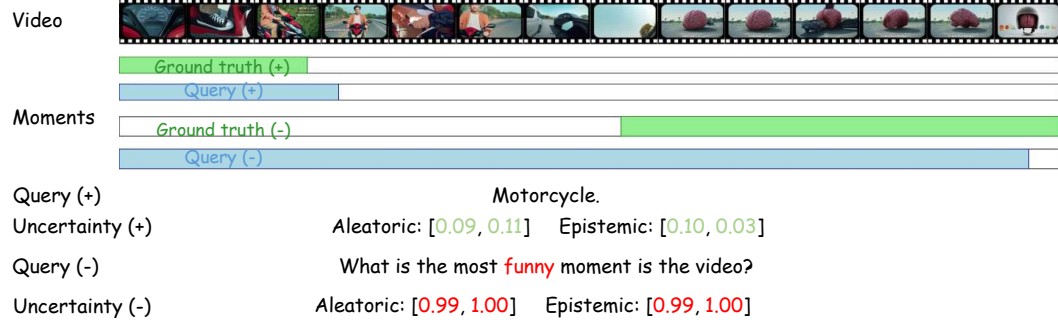

Figure 16: **Adversarial Case IV.** We select an advertisement video containing humor. We provide DDM-VTG with both simple query and abstract query for prediction. Our results showed that DDM-VTG struggle to provide accurate localization for the abstract queries but exhibit high uncertainty.

# E DETAILED WORKFLOW OF THE RFF BLOCK

The RFF block processes inputs from the video and text branches, with initial features denoted as $V^{(1)}$ and $Q^{(1)}$ respectively. The cross-attention module, which shares parameters, alternates the roles of the video and text branches as queries and keys/values:

$$CA^{(1)}_{v \to q} = \text{Softmax} \left( \frac{V^{(1)} Q^{(1)\top}}{\sqrt{d_k}} \right) Q^{(1)} \tag{45}$$

$$CA^{(1)}_{q \to v} = \text{Softmax} \left( \frac{Q^{(1)} V^{(1)\top}}{\sqrt{d_k}} \right) V^{(1)} \tag{46}$$

Following cross-attention, each branch refines its features through self-attention:

$$SA^{(1)}_v = \text{Softmax} \left( \frac{CA^{(1)}_{v \to q} CA^{(1)\top}_{v \to q}}{\sqrt{d_k}} \right) CA^{(1)}_{v \to q} \tag{47}$$

$$SA^{(1)}_q = \text{Softmax} \left( \frac{CA^{(1)}_{q \to v} CA^{(1)\top}_{q \to v}}{\sqrt{d_k}} \right) CA^{(1)}_{q \to v} \tag{48}$$

The outputs of the self-attention stages serve as inputs to the next iteration of the block, thus promoting progressive enhancement of modality alignment. This process continues until the $n$-th layer, after which the refined features of the video and query are output. Therefore, when $1 \leq i \leq n - 1$, the iterative expression is given as follows.

$$V^{(i+1)} = SA^{(i)}_v \tag{49}$$
$$Q^{(i+1)} = SA^{(i)}_q \tag{50}$$

# F DETAILS OF THE VTG HEAD

## F.1 MOMENT RETRIEVAL HEAD

The design of this head is similar to the foreground head, except it features a last layer with two output channels for the left and right offsets. Given $\tilde{\mathbf{V}}^k \in \mathbb{R}^{L_v \times D}$, this head generates a series of offsets $\{\tilde{m}_i\}_{i=1}^{L_v}$ for each unit. We then define the predicted boundary $\tilde{m}_i$ and the corresponding interval $d_i$ (i.e., $d_i = m_i^s - m_i^e$). For training objectives, we use a combination of smooth L1 loss and generalized IoU loss to optimize the model's performance.

$$\mathcal{L}_b = \mathbb{1}_{f_i=1} \left[ \lambda_{L1} \mathcal{L}_{\text{SmoothL1}} \left( \tilde{d}_i, d_i \right) + \lambda_{\text{iou}} \mathcal{L}_{\text{iou}} \left( \tilde{m}_i, m_i \right) \right]. \tag{51}$$

Notably, this regression objective is only devised for foreground clips *i.e.,* $f_i = 1$.

## F.2 VIDEO SUMMARIZATION HEAD

From the frozen video encoder, the output $\tilde{\mathbf{V}}^k \in \mathbb{R}^{L_v \times D}$ passes through a series of three $1 \times 3$ convolutional layers, each layer having $D$ filters and equipped with ReLU activation functions. Following these layers, sigmoid activations are used to generate the prediction $\tilde{f}_i$ for each unit. Focal loss serves as the training objective, with $\gamma = 2.0$ and $\alpha = 0.9$.

$$\mathcal{L}_f = -\lambda_f \alpha (1 - \tilde{f}_i)^\gamma \log(\tilde{f}_i) \tag{52}$$

## F.3 HIGHLIGHT DETECTION HEAD

Given that saliency is defined as the relevance between visual context and a text query, it is appropriate to assess this relationship through a similarity measure between video and text modalities. Let the video tokens be denoted as $\{\mathbf{v}_i\}_{i=1}^{L_v} \in \mathbb{R}^{L_v \times D}$ and the sentence representation as $\mathbf{S} \in \mathbb{R}^{1 \times D}$. We

then calculate the predicted saliency score $\tilde{s}_i$ for each video token $\mathbf{v}_i$ in relation to the text query $Q$, using their cosine similarities.

$$\tilde{s}_i = \cos(\mathbf{v}_i, \mathbf{S}) := \frac{\mathbf{v}_i^T \mathbf{S}}{\|\mathbf{v}_i\|_2 \|\mathbf{S}\|_2}, \tag{53}$$

where $\| \cdot \|_2$ represents the $L2$-norm of a vector.

For each video $\mathbf{V}$, we randomly sample a foreground clip $\mathbf{v}_p$ with $f_p = 1$ and $s_p > 0$ as a positive sample; we treat other clips in the same video $\mathbf{v}_j$ with saliency $s_j$ less than $s_p$ as negative samples, i.e., $\Omega = \{j | s_j < s_p, 1 \leq j \leq L_v\}$, and perform **intra-video** contrastive learning:

$$\mathcal{L}_s^{\text{intra}} = -\log \frac{\exp(\tilde{s}_p / \tau)}{\exp(\tilde{s}_p / \tau) + \sum_{j \in \Omega} \exp(\tilde{s}_j / \tau)}, \tag{54}$$

where $\tau$ is a temperature parameter and set as $0.07$. And we further propose query-driven clip-by-clip contrastive learning where clips within the target moment are treated as positive samples and clips outside as negative samples. Specifically, samples are selected based on the salience scores, with positive samples ranked in descending order and negative samples in ascending order. The top $K$ samples from each are chosen for similarity computation. Given two sets of samples, Pos (positive) and Neg (negative), each containing $K$ elements, the similarity is computed using the dot product, resulting in a similarity matrix $\mathbf{S}$. The similarity matrix $\mathbf{S}$ is derived from the dot product between vectors $\mathbf{v}_i^+$ from the positive set Pos and $\mathbf{v}_j^-$ from the negative set Neg. Each vector represents a moment in the video, with $\mathbf{v}_i^+ \in$ Pos and $\mathbf{v}_j^- \in$ Neg. The similarity $S_{ij}$ between any two moments is computed as follows:

$$S_{ij} = (\mathbf{v}_i^+) \cdot (\mathbf{v}_j^-)^T, \tag{55}$$

The loss function is defined as the negative mean of the trace of $\mathbf{S}$, formally given by:

$$\mathcal{L}_v^{\text{intra}} = -\frac{1}{N} \sum_{i=1}^{N} \mathbf{S}_{ii}. \tag{56}$$

where $N$ is the clip number of the training set. In datasets other than QVHighlight Lei et al. (2021b), where ground truth salience scores are not provided, the foreground flag $f$ is used to dichotomize the samples into positive and negative sets. $K$ samples are then randomly selected from each set for computing the similarity and loss in terms of Eq. 55 and 56.

Besides, we regard sentences from other samples within batches $k \in B$ as negative samples, and develop the **inter-video** contrastive learning for cross-sample supervision:

$$\mathcal{L}_s^{\text{inter}} = -\log \frac{\exp(\tilde{s}_p / \tau)}{\sum_{k \in B} \exp(\tilde{s}_p^k / \tau)}, \tag{57}$$

where $B$ is the training batch size and $\tilde{s}_p^k = \cos(\mathbf{v}_i, \mathbf{S}_k)$.

Our saliency score head training loss is the combination of inter- and intra-video contrastive learning:

$$\mathcal{L}_s = \lambda_{\text{inter}} \mathcal{L}_s^{\text{inter}} + \lambda_{\text{intra}} (\mathcal{L}_s^{\text{intra}} + \mathcal{L}_v^{\text{intra}}). \tag{58}$$

To this end, our grounding objective is the combination of each head loss overall clips in the training set.

$$\mathcal{L}_G = \frac{1}{N} \sum_{i=1}^{N} (\mathcal{L}_f + \mathcal{L}_b + \mathcal{L}_s), \tag{59}$$

where $N$ is the clip number of the training set.

