# OpenReview forum: "Debiased Deep Evidential Regression for Video Temporal Grounding"
_ICLR.cc/2025/Conference — Submitted to ICLR 2025_

### Official Review · Reviewer_yg5S · 2024-10-29

**Soundness:** 2
**Presentation:** 2
**Contribution:** 3
**Rating:** 8
**Confidence:** 3

**Summary:**

This paper studies the issue of open-world challenges caused by open-vocabulary queries and out-of-distribution videos in video temporal grounding. The authors adopt the Deep Evidential Regression as baseline, and propose a Reflective Flipped Fusion block to realize modality alignment and query reconstruction. Meanwhile, a Geom-regularizer is proposed to debias and calibrate uncertainty estimation. Extensive experiments are conducted on the public dataset to validate the proposed method.

**Strengths:**

1. This paper extends the deep evidential regression to video temporal grounding for uncertainty estimation.
2. The authors propose a Geom-regularizer to solve the counterintuitive uncertainty and calibrate the estimation of uncertainty.
3. The proposed method achieves comparable performance in the majority of benchmarks.

**Weaknesses:**

1. The evaluation of location bias is insufficient. There are no transfer experiments on the Charades-CD and ActivityNet-CD datasets to validate the model in OOD scenarios, as done by MomentDETR and MomentDiff.
2. The study of query reconstruction (QR) is not thorough. The authors only present performance across different QR epochs and learning rates.
3. Insufficient performance evaluation. Ego4D-NLQ is widely used in previous works, yet this study does not report results on this dataset. Additionally, the paper fails to compare with recent works, such as "R2-Tuning: Efficient Image-to-Video Transfer Learning for Video Temporal Grounding" from ECCV 2024.

**Questions:**

1. Why does the model only mask and reconstruct one noun? Would masking more words help enhance text sensitivity?
2. In the conclusion, the authors claim that the model’s capabilities are limited by data quality and scale. ActivityNet-Captions and Ego4D-NLQ are large-scale datasets. Would the model perform well on these two datasets?

---

> ### Author Response · Authors · 2024-11-23
> **1. Supplementing the evaluation of location bias (Addressing Weakness 1)**
>
> We have provided additional results for DDM-VTG on the ActivityNet-CD and Charades-CD datasets, including downstream performance and average uncertainty metrics (**see Answer 5 to Reviewer QN6J**). Additionally, we report the average uncertainty of all samples across in-domain (iid) and out-of-domain (ood) test sets for the two CD datasets. The results demonstrate that uncertainty for ood samples is significantly higher than for iid samples, indicating that the uncertainty estimation by DDM-VTG is generally reasonable.
> |  | iid | ood (CD) |
> | --- | --- | --- |
> | ActivityNet-CD | 0.09 | 0.14 |
> | Charades-CD | 0.03 | 0.11 |

---

> ### Author Response · Authors · 2024-11-23
> **2. On the study of the QR task (Addressing Weakness 2)**
>
> The improvements introduced by the QR task to baseline methods, as well as its impact on downstream performance, are presented in Table 3(a) and Figure 6 of the main text. Additionally, we conducted experiments on the effect of `mask_ratio` for QR on the QVHighlight dataset, as shown below:
> | mask_ratio | w/o. mlm | 1 noun | 0.25 | 0.5 | 0.75 | all noun |
> | --- | --- | --- | --- | --- | --- | --- |
> | mAP| 32.75 | **36.93** | 33.40 | 32.43 | 31.80 | 31.71 |
> | R1@0.5 | 57.23 | **64.06** | 61.74 | 59.55 | 59.23 | 58.52 |
> | R1@0.7 | 37.11 | **43.61** | 39.17 | 37.13 | 34.26 | 34.45 |
>
> From these results, we observe that QR affects the model’s ability to align text and video modalities. When the `mask_ratio` is very high, the model struggles to answer QR tasks correctly due to insufficient text context, making it difficult to align video clues. We also experimented with masking nouns, as they carry significant semantic information in VTG tasks. When all nouns are masked, the text-video alignment performance deteriorates significantly. Conversely, masking only one noun enables the model to leverage multi-modal context to achieve better cross-modal alignment.
>
> Notably, while QR’s primary goal is to address modality imbalance in uncertainty estimation, it also positively impacts downstream performance when properly configured.

---

> ### Author Response · Authors · 2024-11-23
> **3. Comparison with R2-Tuning (Addressing Weakness 3)**
>
> R2-Tuning, a concurrent work, was not considered in our initial submission. However, our method demonstrates competitive downstream performance:
>
> | Method | QVHighlight R1@0.5 | QVHighlight R1@0.7 | QVHighlight HD mAP | Charades-STA R1@0.5 | Charades-STA R1@0.7 | Charades-STA mIoU | TACoS R1@0.5 | TACoS R1@0.7 | TACoS mIoU |
> | --- | --- | --- | --- | --- | --- | --- | --- | --- | --- |
> | R2-Tuning | **68.7** | **52.1** | **40.6** | 59.8 | 37.0 | 50.9 | **38.7** | **25.1** | **35.9** |
> | DDM-VTG | 65.0 | 49.4 | 40.1 | **60.2** | **38.0** | **51.6** | 37.3 | 19.4 | 33.9 |

---

> ### Author Response · Authors · 2024-11-23
> **4. On masking a single noun (Addressing Question 1)**
>
> We have experimented with randomly masking arbitrary words, but found this approach to be less efficient than masking only entities. The improvement in performance was slower and, in many cases, negligible. We attribute this to the presence of many words in the text that are irrelevant to the core semantics. **We argue that nouns are the most critical**, primarily because **CLIP features** are more focused on objects. Since CLIP is trained on static images, it emphasizes static visual features and lacks the dynamic visual priors embedded in verbs. Furthermore, in the benchmarks used in this study, successfully identifying the primary nouns is often sufficient for producing high-quality reasoning results. This could explain why masking and reconstructing verbs did not lead to significant improvements.
>
> We believe that reconstructing more types of text, such as verbs (which directly correspond to temporal features), could be beneficial if not constrained by the limitations of the feature extractor. In future work, we plan to explore this hypothesis using more powerful pre-trained feature extractors or large-scale temporal pre-training methods to validate the potential benefits of reconstructing arbitrary words.

---

> ### Author Response · Authors · 2024-11-23
> **5. On the Impact of Data Quality and Scale on Method Performance (Regarding Question 2)**
>
> We believe that the scale of the training dataset plays a crucial role in expanding the knowledge boundaries of the model, thereby improving the robustness of its uncertainty estimation. To evaluate this, we conducted experiments using 20%, 40%, 60%, 80%, and 100% of the training splits. We report both the VTG task performance and the uncertainty quantification metrics of the model outputs. As shown in the table below, increasing the data scale leads to consistent improvements in both VTG task performance and uncertainty quantification ability.
>
> To assess the impact of data quality, we designed an adversarial experiment where the quality of data was intentionally degraded during inference. The results demonstrate that the model outputs higher uncertainty when encountering low-quality or abnormal data, confirming its ability to quantify uncertainty effectively. Details of this experiment can be found in **Answer 3 to Reviewer MSwQ**, including the cases presented in **Figure 11**, **Appendix C.3**, and **Figures 13–16**.
>
> | **Training Set Scale** | **QVHighlights VTG Performance** |  | **QVHighlights Uncertainty** |  | **Charades-STA VTG Performance** |  | **Charades-STA Uncertainty** |  |
> | --- | --- | --- | --- | --- | --- | --- | --- | --- |
> |  | **mAP-MR** | **Hit1-HL** | **EUCM↓** | **Entropy↑** | **mAP-MR** |  | **EUCM↓** | **Entropy↑** |
> | **20%** | 25.67 | 57.94 | 0.3681 | 0.2649 | 33.02 |  | 0.1639 | 0.0987 |
> | **40%** | 33.60 | 61.42 | 0.2813 | 0.1795 | 37.05 |  | 0.1620 | 0.2216 |
> | **60%** | 37.25 | 63.16 | 0.2805 | 0.2388 | 39.68 |  | 0.1444 | 0.3479 |
> | **80%** | 38.82 | **64.06** | 0.3226 | **0.2983** | 40.15 |  | 0.1596 | 0.2919 |
> | **100%** | **41.15** | 64.00 | **0.2787** | 0.2457 | **41.43** |  | **0.1440** | **0.3609** |
> ---
> Also, we appreciate the reviewer’s suggestion and have further supplemented our analysis with additional results on the **ActivityNet-Captions** and **Ego4D-NLQ** datasets. Since several prominent baselines do not report results on these datasets, we included comparisons with additional models. The results indicate:
>
> 1. Our method demonstrates strong downstream task performance on both large-scale datasets.
> 2. Based on these findings, we plan to further explore larger-scale pretraining to improve both uncertainty estimation and downstream task performance.
>
> | **Dataset** | **Method** | **R1@0.3** | **R1@0.5** | **R1@0.7** |
> | --- | --- | --- | --- | --- |
> | **ActivityNet-Captions** | VLG-NET [1] | 46.32 | 29.82 | - |
> |  | UnLoc-large [2] | 48.30 | 30.20 | - |
> |  | **Ours** | **71.72** | **56.34** | **33.68** |
> | **Ego4D-NLQ** | UniVTG [3] | 7.28 | 3.95 | 1.32 |
> |  | EgoVLP [4] | 10.46 | 6.24 | - |
> |  | **Ours** | **11.04** | **8.18** | **5.32** |
>
> [1] Soldan, Mattia, et al. VLG-Net: Video-Language Graph Matching Network for Video Grounding. ICCVW'21.
>
> [2] Yan, Shen, et al. Unloc: A unified framework for video localization tasks. ICCV'23.
>
> [3] Lin, Kevin Qinghong, et al. Univtg: Towards unified video-language temporal grounding. ICCV'23.
>
> [4] Lin, Kevin Qinghong, et al. Egocentric video-language pretraining. NeurIPS'22.

---

> > ### Comment · Reviewer_yg5S · 2024-11-24
> >
> > Thank you for your response. My concerns have been addressed. I am willing to raise my rating.

---

> > > ### Author Response · Authors · 2024-11-24
> > >
> > > Dear Reviewer,
> > >
> > > We are grateful for your positive update. We sincerely appreciate your constructive feedback and valuable advice!

---

> ### Author Response · Authors · 2024-11-25
>
> Dear reviewer,
>
> Thanks again for your positive response. However, we noticed that the score has not yet been updated, so we wanted to kindly check if there is any issue or if additional actions are required on our end to facilitate the update.

---

### Official Review · Reviewer_FPPU · 2024-11-02

**Soundness:** 2
**Presentation:** 3
**Contribution:** 3
**Rating:** 5
**Confidence:** 4

**Summary:**

This paper proposes Debiased DER Model for VTG, tackling open-vocabulary queries and out-of-distribution videos in video temporal grounding tasks. It extends the vanilla DER to VTG and establishes a baseline. To address two critical biases in the baseline—modality imbalance and counterintuitive uncertainty—the method incorporates a RFF block for progressively enhancing modal alignment, a query reconstruction task to ensure robust cross-modal alignment capabilities and a Geom-regularizer to calibrate uncertainty estimation. The proposed method has been evaluated on 4 datasets, demonstrating its effectiveness in Moment Retrieval, Highlight Detection and Video Summarization. The ablation studies also support the analysis.

**Strengths:**

- The basic idea is easy to follow and the main motivation is clear.
- The innovative integration of DER into VTG tasks is a novel approach that effectively addresses key issues like OOD videos.
- The proposed method achieves strong experiment results, both compared to its baseline and other SOTA methods.

**Weaknesses:**

- In figure3, I can’t see the difference between the two distributions except for the color, which might be confusing as to why one is unreliable and the other is trustworthy.
- About the presentation. In 4.3, there is a significant disparity in the level of detail explained for different modules, perhaps the arrangement of content in the main text and appendix could be adjusted to make it clearer for readers.
- The experimental section only shows the comparison with SOTA methods on various metrics. In the appendix, only some cases of the QVHighlights dataset are shown, without visual results for the other datasets mentioned in the paper, and it also lacks displays of comparative results for the three sub-tasks.
- It would be more complete to have a discussion of this increased cost if there are any, as well as techniques used to overcome it.
- (Minor) Minor typos/grammatical mistakes (e.g. 4.2 “VALLINA”)

**Questions:**

- In Figure 2, several challenges within VTG tasks are highlighted, but it appears that targeted comparative experiments were not conducted in the study. When compared with other works, can DDM-VTG perform better in addressing these challenges? Some discussions are expected.
- In the Query Reconstruction task, how can DDM-VTG ensure that the tokens predicted by the QR head are accurate when dealing with complex videos? What happens if the predictions are incorrect? Does it affect the accuracy of temporal localization of the whole video?
- In the case study, the average length of the videos is 150 seconds. How would the model perform with longer videos, and would the cost increase significantly?

---

> ### Author Response · Authors · 2024-11-22
> **1. Regarding the readability of Figure 3 (Addressing Weakness 1)**
>
> To improve readability and better highlight the main innovation of the Geom-Regularizer in uncertainty modeling, we are revising Figure 3 to clarify the distinction between different components. The updated version will be included in the final manuscript.

---

> ### Author Response · Authors · 2024-11-22
> **2. On content distribution in the main text (Addressing Weakness 2)**
>
> Considering the page limit, we slightly reduced the description of the model architecture in the main text. We will optimize the layout and provide further refinements later.

---

> ### Author Response · Authors · 2024-11-22
> **3. Additional datasets and visual results (Addressing Weakness 3)**
>
> Cases on additional datasets will be uploaded to the supplementary material (**rebuttal.zip**). Results on other sub-tasks have been provided in **Table 1&2** of the main text.

---

> ### Author Response · Authors · 2024-11-22
> **4. On computational cost (Addressing Weakness 4)**
>
> Our method introduces minimal additional computational overhead. Specifically:
>
>    - The inclusion of the RFF block, QR task, and DER with the Geom-Regularizer into the DDM-VTG framework has negligible runtime impact. Empirical measurements on an NVIDIA Tesla V100 GPU show consistency with the baseline model.
>    - From a theoretical perspective, the training process for DDM-VTG does not incur significant computational costs. As shown in Appendix A.2, the NLL loss for DER is expressed as:
>
> $$
> \begin{aligned} \mathcal{L}^{\text{NLL}}_i &= \frac{1}{2} \log \left( \frac{\pi}{\nu} \right) - \alpha \log(\Omega) + \left( \alpha + \frac{1}{2} \right) \log \left( (b_i - \gamma)^2 \nu + \Omega \right) + \log \left( \frac{\Gamma(\alpha)}{\Gamma \left( \alpha + \frac{1}{2} \right)} \right) \end{aligned}
> $$
>
>    - Each term in the NLL loss has a temporal complexity of O(1). For a training set of N samples, computing the loss for both left and right boundaries results in a total complexity of O(N).
>    - Similarly, the spatial complexity of the Geom-Regularizer is O(1), as it does not require additional storage. Hence, the overall complexity of our proposed DER + Geom-Regularizer framework is \(O(N)\), consistent with the baseline model.

---

> ### Author Response · Authors · 2024-11-22
> **5. On spelling and grammatical errors (Addressing Weakness 5)**
>
> We sincerely thank the reviewers for their careful reading. All identified spelling and grammatical errors will be corrected in the final submission.

---

> ### Author Response · Authors · 2024-11-22
> **6. On addressing the challenges shown in Figure 2 (Addressing Question 1)**
>
> Existing VTG methods lack any explicit mechanisms to quantify uncertainty. As pioneers in this field, we established a baseline and, for the first time, achieved explicit uncertainty quantification. We further proposed DDM-VTG to debias this uncertainty quantification.
>
> Quantitative and qualitative results related to Figure 2 are provided in the main text (as noted in **Answer 3 to Reviewer MSwQ**). In addition, we have included quantitative results on the ActivityNet-CD and Charades-CD datasets to address the challenges outlined in Figure 2(a), as referenced in **Answer 5 to Reviewer QN6J** . Visual comparisons on additional datasets will be included in the supplementary material (**rebuttal.zip**).

---

> > ### Comment · Reviewer_FPPU · 2024-11-28
> >
> > There is still a lack of analytical and exploratory discussion regarding the experimental results.

---

> ### Author Response · Authors · 2024-11-22
> **7. On the impact of QR on downstream performance (Addressing Question 2)**
>
> The QR task is an auxiliary task designed to enhance text-video modality understanding during training and mitigate the modality imbalance observed in baseline methods when estimating uncertainty. Inspired by "mask-and-reconstruct" approaches like MAE and BERT, the QR task masks parts of the input text and reconstructs it during training. However, during inference, the QR head is not used for text reconstruction. Instead, the model is provided with the full text, so there is no issue of "incorrect reconstructed query" during inference.
>
> The improvements introduced by the QR task to baseline methods, as well as the accompanying downstream performance gains, are presented in Table 3(a) and Figure 6 of the main text. Additionally, we have conducted experiments on the effect of the `mask_ratio` for QR on the QVHighlight dataset, and you can refer to Answer 2 to Reviewer yg5S.

---

> > ### Comment · Reviewer_FPPU · 2024-11-28
> >
> > Thanks for your reply. This question has been addressed.

---

> ### Author Response · Authors · 2024-11-22
> **8. On inference for longer videos (Addressing Question 3)**
>
> The applicability of our proposed uncertainty estimation method to longer videos is inherently linked to the length of videos seen during training. If the model is trained on datasets with long videos, it is reasonable to believe that the model has the potential to generalize to even longer videos.
>
> As for the computational cost, it is primarily determined by the backbone architecture. Since our focus is on achieving reliable uncertainty estimation, we maintain consistency with existing VTG methods in feature extraction. Thus, the computational complexity is largely driven by the quadratic time complexity of the transformer, which makes inference on long videos inherently resource-intensive.
>
> However, with the continuous advancements in long video research, we are optimistic that our proposed method can be scaled up to support long video inference with reduced computational cost in the future.

---

> > ### Comment · Reviewer_FPPU · 2024-11-28
> >
> > Due to the lack of substantial theoretical and experimental analysis, with only intuitive and vague responses, this question has not been adequately addressed.

---

> ### Author Response · Authors · 2024-11-24
>
> Dear reviewer,
>
> We wonder if our response answers your questions and addresses your concerns? If yes, would you kindly consider raising the score? Thanks again for your very constructive and insightful feedback!

---

> ### Author Response · Authors · 2024-11-25
>
> Dear reviewer:
>
> With the discussion stage ending soon, we wanted to kindly follow up to check if our response has addressed your questions and concerns. If yes, would you kindly consider raising the score before the discussion phase ends？ We are truly grateful for your time and effort！

---

> ### Author Response · Authors · 2024-12-02
> **On addressing the challenges shown in Figure 2 (Addressing Question 1)**
>
> Thanks for your point. In response, we have carefully revisited all the relevant experimental results related to Figure 2 and reorganized our discussion to provide a clearer and more thorough analysis.
>
> 1. **Addressing the Challenges in Figure 2:**
> We have conducted a series of quantitative experiments involving VTG downstream tasks (Tables 1, 2, and additional results, such as Answer 5 to **Reviewer yg5S** comparing performance on large-scale datasets). Our experimental findings show that the proposed method achieves competitive performance relative to recent state-of-the-art VTG methods. This suggests that the uncertainty measurement approach itself is meaningful. While the primary focus of this study is on measuring uncertainty in VTG inference, we are pleasantly surprised to observe that our method also leads to improvements in VTG task performance. We believe this enhancement stems from the fact that modeling uncertainty strengthens the model’s ability to extract important information from the video.
> 2. **Evaluating Uncertainty Measurement Across Modalities:**
> To further validate that our uncertainty measurement method can effectively capture uncertainty introduced by the text and image modalities, we conducted adversarial experiments. By introducing varying levels of noise into the text and image modalities, we observed the resulting changes in uncertainty during inference. This setup directly simulates the challenges described in Figure 2. The uncertainty distributions (shown in Figure 6) indicate that our method is highly sensitive to adversarial samples in both modalities, which helps address the modality imbalance issue present in the baseline model.
> 3. **Addressing the Issue of Video Length Distribution:**
> As noted, videos of varying lengths are common in real-world applications. Reference [1] discusses how current VTG datasets are affected by the uneven distribution of video lengths, leading to potential bias since models often focus on specific time intervals where most localization occurs. To address this, we visualized the uncertainty inference results on QVHighlight (Figure 8) and conducted ablation studies. The results demonstrate that our method can effectively capture this issue through uncertainty measurement, with the model exhibiting significantly higher uncertainty for less frequent localization results. This further highlights that DDM-VTG exhibits strong capabilities in handling out-of-distribution (OOD) samples and mitigating location bias.
>
>     In response to the concerns raised by **Reviewer QN6J** and **Reviewer yg5S** regarding the generalization of DDM-VTG, we have added experiments on ActivityNet-CD and Charades-CD as suggested (please refer to Answer 5 to Reviewer QN6J). In these datasets, the IID samples represent data processed to address location bias, while the OOD samples are unprocessed. The results show that our method outperforms the baseline in OOD inference and achieves a smaller **IID-OOD Gap**, indicating that DDM-VTG is robust to location bias.
>
> 4. **Qualitative Case Studies:**
> While we have presented numerous quantitative results, our primary goal is to demonstrate that the model can make robust inferences even when confronted with clearly anomalous inputs, as outlined in Figure 1. To visually demonstrate this ability, we have included a series of case studies (**Figure 11 in Appendix C.3**, and **Figures 13–16 in Appendix C.5**). These cases cover the challenges mentioned in Figure 2(a), (b), and (d). Since existing VTG methods do not explicitly quantify uncertainty to address these challenges, we chose UniVTG as a representative baseline for comparison. The case-by-case comparisons and analyses have been provided in the **rebuttal.zip**. These examples confirm that our model is able to sensibly and accurately express the uncertainty involved in inferences when facing these challenges.
>
>     However, due to the inherent limitations of the model's knowledge capacity, we observed that the model's inference on extreme samples still lacks fine-grained characterization. For example, similar extreme OOD samples tend to consistently show extreme uncertainty values, such as 0.99 or 1.00. We sincerely hope to further improve this aspect in future work, allowing the model to produce more nuanced and reliable uncertainty estimates, which would lead to more trustworthy VTG inferences.
>
>
> [1] Yuan, Yitian, et al. A closer look at temporal sentence grounding in videos: Dataset and metric. *Proceedings of the 2nd international workshop on human-centric multimedia analysis*. 2021.

---

> ### Author Response · Authors · 2024-12-02
> **On inference for longer videos (Addressing Question 3)**
>
> To the best of our knowledge, there is no existing benchmark specifically designed to evaluate the performance of VTG models on test sets with varying video lengths, particularly for longer videos. To address this, we created our own benchmark based on QVHighlight: We randomly select `x` videos from  the corresponding splits and concatenate with the original video with a random order, where `x` takes value from 0 to 5. Which means we create 6 datasets (with 6 length levels ), from 150 seconds to 15 minutes.  We controlled for the same number of training epochs (epoch = 180) to obtain the experimental results and training durations. And 4 Tesla V100-32G are applied.
>
> |  | x=0 | x=1 | x=2 | x=3 | x=4 | x=5 |
> | --- | --- | --- | --- | --- | --- | --- |
> | MR-R1@0.3 | 76.45 | 71.94 | 68.90 | 67.74 | 65.94 | 59.61 |
> | MR-R1@0.5 | 64.77 | 59.81 | 57.42 | 55.48 | 55.35 | 46.06 |
> | MR-R1@0.7 | 45.68 | 38.58 | 36.77 | 35.74 | 36.77 | 27.10 |
> | MR-R5@0.3 | 90.65 | 87.29 | 86.32 | 84.45 | 83.94 | 77.23 |
> | MR-mAP | 40.09 | 35.10 | 33.20 | 32.54 | 33.54 | 26.68 |
> | HL-mAP | 39.85 | 38.56 | 26.8 | 19.82 | 16.57 | 19.83 |
> | HL-Hit1 | 63.74 | 61.48 | 41.29 | 28.52 | 23.16 | 22.84 |
> | Training Time | t | 1.13t | 1.18t | 1.23t | 1.31t | 1.46t |
>
> The results indicate that as the video length increases, the model's downstream performance gradually deteriorates under the same number of epochs. This degradation is likely related to underfitting and the increased complexity of the contextual information that the model needs to process. However, we observed that the time required to run the model did not increase significantly when using the same number of epochs. Specifically, for x=5 (15-minute, micro-film-length videos), the runtime was only 1.46 times longer than that for x=0, with the same number of epochs.
>
> Additionally, we included zero-shot inference results of models trained on the original video length, directly applied to the constructed long video dataset’s val splits. And only a single Tesla V100-32G is applied.
>
> |  | x=0 | x=1 | x=2 | x=3 | x=4 | x=5 |
> | --- | --- | --- | --- | --- | --- | --- |
> | MR-R1@0.3 | 76.45 | 67.94 | 61.35 | 51.35 | 41.03 | 35.29 |
> | MR-R1@0.5 | 64.77 | 50.77 | 36.13 | 24.97 | 18.00 | 13.10 |
> | MR-R1@0.7 | 45.68 | 26.58 | 15.35 | 10.05 | 7.83 | 5.03 |
> | MR-R5@0.3 | 90.65 | 71.68 | 65.87 | 56.97 | 46.84 | 40.13 |
> | MR-mAP | 40.09 | 23.96 | 15.53 | 10.55 | 7.23 | 5.36 |
> | HL-mAP | 39.85 | 37.49 | 26.47 | 19.29 | 15.84 | 13.51 |
> | HL-Hit1 | 63.74 | 68.90 | 41.48 | 27.55 | 23.23 | 18.90 |
> | Inference Time | t | 1.84t | 2.54t | 3.31t | 4.23t | 4.61t |
>
> We can observe that the model is affected to some extent by the more complex contextual information and Temporal OOD. In the case of zero-shot inference, its performance significantly degrades when applied to `x`=3, 4, and 5, which have longer video lengths. This suggests that directly applying a model trained on shorter videos (150s) to inference on longer videos results in noticeable downstream performance degradation, with this degradation becoming more pronounced as the video length increases. However, we observed that the inference time generally increases linearly with video length. Specifically, when performing inference on a 900s video, the inference time is 4.61 times longer than for a 150s video, which seems reasonable.

---

> > ### Author Response · Authors · 2024-12-03
> >
> > Dear reviewer FPPU:
> >
> > With the discussion stage ending soon, we want to kindly follow up to check if our response has addressed your questions and concerns. If yes, would you kindly consider raising the score before the discussion phase ends？ We are very grateful for your time and effort！

---

### Official Review · Reviewer_QN6J · 2024-11-03

**Soundness:** 3
**Presentation:** 3
**Contribution:** 3
**Rating:** 6
**Confidence:** 3

**Summary:**

The paper presents a novel approach to Video Temporal Grounding (VTG) by integrating Deep Evidential Regression (DER) to address uncertainties in open-world scenarios, such as out-of-distribution (OOD) data and open-vocabulary queries. The authors propose a Debiased DER Model for Video Temporal Grounding (DDM-VTG) that tackles modality imbalance and counterintuitive uncertainty through a Reflective Flipped Fusion (RFF) block, a query reconstruction task, and a Geom-regularizer. The model demonstrates effectiveness and robustness across multiple benchmarks.

**Strengths:**

1.	The proposed baseline model is innovative for its integration of Deep Evidential Regression (DER) with VTG tasks to address both aleatoric and epistemic uncertainties.
2.	The paper not only identifies the existence of modal imbalance and structural flaws in regularization within the baseline model but also offers solutions to these issues.
3.	The authors have conducted extensive experiments across various benchmarks, which effectively demonstrate the efficacy of their approach.

**Weaknesses:**

1.	While the paper presents a novel approach to addressing uncertainties in VTG, it could benefit from a deeper analysis of the limitations of the proposed model, especially in handling highly ambiguous queries or extremely OOD data.
2.	The paper could provide more insights into how the DDM-VTG model generalizes to other video-related tasks beyond the tested benchmarks.
3.	When designing the baseline, whether DER provides positive assistance for the correct prediction of the model, the author needs to provide corresponding proof experiments.
4.	When introducing the baseline, the author believes that it has a modal imbalance problem, and DDM-VTG effectively alleviates this imbalance, which requires corresponding experimental evidence.
5.	The method proposed by the author showed out of distribution predictions on the qv height dataset, which to some extent indicates the generalization of DDM-VTG, but it is not clear and specific enough. The author needs to provide results on charades-CD.

**Questions:**

Please see the weaknesses.

---

> ### Author Response · Authors · 2024-11-22
> **1. Discussion on the Limitations of DDM-VTG**
>
> Due to model knowledge capacity and dataset constraints, DDM-VTG is unable to provide more fine-grained uncertainty estimates for extreme out-of-distribution (OOD) samples. Specifically, in our experiments, we observed that the uncertainty distribution estimated by DDM-VTG for varying degrees of OOD samples is not sufficiently uniform. For example, similar extreme OOD samples are consistently estimated with extreme uncertainty values such as 0.99 or 1.00. We plan to address this issue in future work through targeted optimization.

---

> > ### Author Response · Authors · 2024-11-22
> > **2. On the Generalization of DDM-VTG to Other Video-Related Tasks**
> >
> > In Tables 1 and 2, we present the evaluation results of DDM-VTG on three video-related tasks: Moment Retrieval, Highlight Detection, and Video Summarization. Our work primarily focuses on uncertainty estimation for **multi-modal regression tasks**, and we believe that DDM-VTG has potential for extension to other multi-modal regression tasks as well.

---

> > > ### Author Response · Authors · 2024-11-22
> > > **3. On Whether DER Corrects Model Predictions**
> > >
> > > We introduce DER to **provide reliable uncertainty estimation in VTG**, though exploring how the model leverages this uncertainty to refine its predictions is not the primary focus of our work. Interestingly, our proposed approach yields notable downstream performance improvements as a serendipitous benefit. These enhancements complement the performance comparison between DDM-VTG and the baseline on downstream tasks, as summarized in Table 1 of the original manuscript. Below is the performance comparison across QVHighlight and Charades datasets:
> > > | **Method**         | **QVH-MR R@0.5** | **QVH-MR R@0.7** | **QVH-MR Avg.M** | **QVH-HD MAP** | **QVH-HD HIT@1** | **Charades-STA R@0.5** | **Charades-STA R@0.7** | **Charades-STA mIoU** |
> > > |:-------------------:|:----------------:|:----------------:|:----------------:|:--------------:|:-----------------:|:----------------------:|:----------------------:|:---------------------:|
> > > | **Baseline (ours)** |       56.8       |       39.2       |       35.3       |      39.8      |       62.2        |         54.7           |         35.4           |         48.6          |
> > > | **DDM-VTG (ours)**  |     **65.0**     |     **49.4**     |     **43.0**     |    **40.1**    |     **63.4**      |       **60.2**         |       **38.0**         |       **51.6**        |

---

> ### Author Response · Authors · 2024-11-22
> **4. On Modality Imbalance**
>
> In Table 3(a), we report ablation experiments demonstrating the effectiveness of our method in addressing modality imbalance by introducing metrics such as $\mathrm{Var_{vis}}$ and $\mathrm{Var_{text}}$. We provide detailed explanations of these metrics and their purposes in lines 386-389 and 410-418, along with an analysis of the ablation results. Additionally, Figure 6 offers empirical evidence of DDM-VTG's ability to alleviate modality imbalance, as discussed in lines 452-470.

---

> ### Author Response · Authors · 2024-11-22
> **5. On Performance on Charades-CD and ActivityNet-CD**
>
> We supplement the evaluation of **DDM-VTG** on Charades-CD and ActivityNet-CD (ANet-CD), reporting both downstream performance and metrics such as average uncertainty. The results are summarized below:
>
> | Method | Charades-CD R1@0.3 | Charades-CD R1@0.5 | Charades-CD R1@0.7 | Charades-CD MAP_avg | ANet-CD R1@0.3 | ANet-CD R1@0.5 | ANet-CD R1@0.7 | ANet-CD MAP_avg |
> | --- | --- | --- | --- | --- | --- | --- | --- | --- |
> | **MMIN (iid)** | - | - | - | - | - | - | - | - |
> | **MMIN (ood)** | 55.91 | 34.56 | 15.84 | 15.73 | 44.13 | 24.69 | 12.22 | 15.06 |
> | ***Δ ↓*** | ***-*** | ***-*** | ***-*** | ***-*** | ***-*** | ***-*** | ***-*** | ***-*** |
> | **MomentDETR (iid)** | - | - | - | - | - | - | - | - |
> | **MomentDETR (ood)** | 57.34 | 41.18 | 19.31 | 18.95 | 39.98 | 21.30 | 10.58 | 12.19 |
> | ***Δ ↓*** | ***-*** | ***-*** | ***-*** | ***-*** | ***-*** | ***-*** | ***-*** | ***-*** |
> | **CM-NAT[1] (iid)** | 64.21 | 53.82 | 34.47 | - | 49.91 | 41.67 | 28.82 | - |
> | **CM-NAT[1] (ood)** | 52.21 | 39.86 | 21.38 | - | 32.32 | 20.78 | 11.03 | - |
> | ***Δ ↓*** | ***12.00*** | ***13.96*** | ***13.09*** | ***-*** | ***17.59*** | ***20.89*** | ***17.79*** | ***-*** |
> | **MomentDiff (iid)** | - | - | - | - | - | - | - | - |
> | **MomentDiff (ood)** | 67.73 | 47.17 | 22.98 | 22.76 | 45.54 | 26.96 | 13.69 | 16.38 |
> | ***Δ ↓*** | ***-*** | ***-*** | ***-*** | ***-*** | ***-*** | ***-*** | ***-*** | ***-*** |
> | **Ours (iid)** | 71.10 | 62.20 | 43.29 | 43.41 | 56.33 | 41.77 | 27.47 | 27.68 |
> | **Ours (ood)** | 67.81 | 52.46 | 30.97 | 32.80 | 41.64 | 23.76 | 16.89 | 17.49 |
> | ***Δ ↓*** | ***3.29*** | ***9.74*** | ***12.32*** | ***10.61*** | ***14.69*** | ***18.01*** | ***10.58*** | ***10.19*** |
> ---
> ### Key Insights:
>
> 1. **Superior OOD Performance:**
>
>     Our method demonstrates strong generalization in out-of-distribution (OOD) settings. For example, on Charades-CD, DDM-VTG achieves a **10.04% higher mAP** than MomentDiff and **13.85% higher mAP** than MomentDETR. This performance highlights DDM-VTG's robustness under distributional shifts.
>
> 2. **Small IID-OOD Gap:**
>
>     The performance gap between IID and OOD splits for DDM-VTG is significantly smaller than that of CM-NAT (e.g., ***3.29% vs. 12.00%*** at Charades-CD R1@0.3). This suggests that DDM-VTG effectively learns generalized features that reduce susceptibility to dataset biases.
>
> 3. **Generalization to Challenging Scenarios:**
>
>     The reduced IID-OOD performance gap, coupled with competitive results across all metrics, underscores DDM-VTG's ability to combat dataset biases and adapt to OOD scenarios more effectively than existing methods.
>
> [1] Lan, Xiaohan, et al. "Curriculum multi-negative augmentation for debiased video grounding." AAAI 2023.

---

> ### Author Response · Authors · 2024-11-24
>
> Dear reviewer,
>
> We wonder if our response answers your questions and addresses your concerns? If yes, would you kindly consider raising the score? Thanks again for your very constructive and insightful feedback!

---

> ### Author Response · Authors · 2024-11-25
>
> Dear reviewer:
>
> With the discussion stage ending soon, we wanted to kindly follow up to check if our response has addressed your questions and concerns. If yes, would you kindly consider raising the score before the discussion phase ends？ We are truly grateful for your time and effort！

---

> ### Author Response · Authors · 2024-12-02
>
> Dear reviewer,
>
> The discussion stage, which has been extended by the ICLR committee, is ending soon, we wonder if our response answers your questions and addresses your concerns? If yes, would you kindly consider raising the score? Thanks again for your very constructive and insightful feedback!

---

> > ### Comment · Reviewer_QN6J · 2024-12-03
> >
> > Thank you for your response, which addresses some of my questions. I have increased my score to 6.

---

> ### Author Response · Authors · 2024-12-03
>
> Dear Reviewer QN6J,
>
> Thank you for the positive update. We greatly appreciate your time and consideration. If there are any remaining concerns that you would like further clarification on before the discussion stage ends, please feel free to let us know. We would be more than happy to address any further questions.

---

### Official Review · Reviewer_MSwQ · 2024-11-04

**Soundness:** 3
**Presentation:** 3
**Contribution:** 3
**Rating:** 5
**Confidence:** 3

**Summary:**

It presents DDM-VTG, a new model that integrates Deep Evidential Regression into Video Temporal Grounding to handle uncertainties in open-world scenarios. It addresses modality imbalance and counterintuitive uncertainty with a Reflective Flipped Fusion block and a Geom-regularizer, enhancing model robustness and effectiveness across benchmarks.

**Strengths:**

1. It introduces the first extension of Deep Evidential Regression (DER) to Video Temporal Grounding (VTG) tasks, aiming to address uncertainties in open-world scenarios.
2. It proposes a Debiased DER Model (DDM-VTG) that tackles modality imbalance and counterintuitive uncertainty through a Reflective Flipped Fusion block and a Geom-regularizer, enhancing the model's sensitivity to text queries and calibrating uncertainty estimation.

**Weaknesses:**

1. The datasets used are not open-world.
2. The performance on the video summarization task is not advantageous enough.
3. Figure 2 shows 4 cases of the uncertainty. It is not clear how the method addresses (a)(b)(d) and how to evaluate if the methods can handle these scenarios.

**Questions:**

1. Since the datasets have annotations like a matched moment and text, how to evaluate the model's ability to learn uncertainty when processing an unreasonable text query? Like the example in Figure 1
2. In Geom-regularization, how to define accurate predictions? how to define less accurate predictions?

---

> ### Author Response · Authors · 2024-11-20
> **1. On Whether the Dataset is Open-World (Addressing Weakness 1)**
>
> We argue that the datasets used in our work are open-world in nature. The majority of our evaluations were conducted on QVHighlights, which employs free-text annotations for video segments rather than restricting to a finite set of categories. Moreover, QVHighlights encompasses diverse video types, including news and vlogs, with no constraints on objects, environments, actions, or domains. Thus, we consider QVHighlights a quintessential open-world dataset, and we performed extensive and comprehensive experiments on it.
>
> Additionally, our experiments included other datasets, such as TACoS and Charades-STA. While these datasets impose certain domain-specific restrictions (e.g., TACoS focuses on kitchen scenarios and Charades-STA on indoor activities), they still exhibit open-world characteristics in other aspects, such as free-text annotations, unconstrained objects, and actions. Therefore, despite some domain limitations, we believe these datasets maintain an open-world nature.

---

> ### Author Response · Authors · 2024-11-20
> **2. On the Performance in Video Summarization (Addressing Weakness 2)**
>
> We acknowledge that the results reported in Table 2 for video summarization, while decent, may appear less competitive due to the following reasons:
>
> 1. **Domain-Specific Challenges in TVSum**
>
>     TVSum includes videos from 10 distinct domains. Although often treated as "a single dataset," prior works have adopted a domain-specific tuning strategy to optimize results separately for each domain. For example, UniVTG[1] explicitly states in Table 1 of its appendix that hyperparameters were chosen via a "search" process. From our observations, using different hyperparameters for different domains can significantly impact performance. While we could employ similar domain-specific optimizations to achieve higher results, we chose to treat TVSum as a single dataset with a shared hyperparameter set to derive a **meaningful average metric**. As shown in Table 2 (Column 11), this approach achieves a 3.6% improvement over UniVTG, which employs domain-specific tuning.
>
> 2. **Focus on Uncertainty Estimation, Not Downstream Tasks**
>
>     The primary focus of this work is improving uncertainty estimation in VTG through innovations such as the RFF block, QR task, and geom-regularizer. The downstream task results reported in Table 2 aim to demonstrate that our approach provides solid multimodal understanding and VTG capabilities, ensuring the reliability of our core uncertainty-related discussions. While downstream performance improvements are a welcome byproduct of these innovations, they are not the central goal of our work.
>
> [1] Lin K Q, Zhang P, Chen J, et al. UniVTG: Towards unified video-language temporal grounding, ICCV. 2023

---

> ### Author Response · Authors · 2024-11-20
> **3. On How DDM-VTG Addresses the Scenarios in Figure 2 and Evaluates Its Effectiveness (Addressing Weakness 3)**
>
> Figure 2 illustrates examples corresponding to the two types of uncertainty modeled by our approach: **epistemic uncertainty** (Figures 2a, 2b) and **aleatoric uncertainty** (Figures 2c, 2d). Our baseline framework already incorporates the modeling and measurement of these uncertainties, as detailed in Sections 3 and 4.2. Building on this, DDM-VTG strengthens modality interaction and uncertainty correction, enabling more robust uncertainty estimation in the challenging multimodal context of VTG tasks.
>
> **We provide comprehensive qualitative and quantitative experiments to demonstrate DDM-VTG’s capability in addressing these scenarios.**
>
> ---
>
> ### **Qualitative Experiments**
>
> We conduct extensive case studies to illustrate DDM-VTG's ability to estimate uncertainties effectively. These cases are presented in **Figure 11**, **Appendix C.3**, and **Figures 13–16 (Appendix C.5)**. Specifically:
>
> - **Figure 2a (OOD Video):**
>     - Case 2 in Figure 11: OOD video with abnormal aspect ratios (lines 1089–1095).
>     - Case 4 in Figure 11: Rare boundary annotations (lines 1103–1109).
>     - Figures 14–15: OOD video domains (lines 1190–1220).
> - **Figure 2b (Semantic Ambiguity):**
>     - Case 6 in Figure 11: Visual blur caused by a "plastic container" (lines 1117–1122).
>     - Figure 13: Tiny objects causing visual blur (lines 1162–1176).
>     - Figure 16: Abstract textual expressions causing semantic ambiguity (lines 1225–1239).
> - **Figure 2d (Low-Level Feature Variations):**
>     - Case 3 in Figure 11: Scene transitions (lines 1096–1102).
>     - Case 5 in Figure 11: Lighting condition changes (lines 1110–1116).
>
> ---
>
> ### **Quantitative Experiments**
>
> We simulate the scenarios in Figure 2 using controlled perturbations and measure the resulting uncertainty distributions on QVHighlights:
>
> - **Simulating OOD and Ambiguities:**
>     - Figure 6 shows how uncertainty distributions change when varying levels of Gaussian noise are added to videos (simulating OOD scenarios in Figure 2a, visual semantic ambiguity in Figure 2b, and low-level feature variations in Figure 2d) or replacing different proportions of tokens in text (simulating text semantic ambiguity in Figure 2b).
>
>
> **These experiments, both qualitative and quantitative, validate DDM-VTG’s capability to robustly address the diverse uncertainty scenarios outlined in Figure 2.**

---

> > ### Author Response · Authors · 2024-11-20
> > **4. On DDM-VTG’s Uncertainty Estimation for Unreasonable Text Queries (Addressing Question 1)**
> >
> > We evaluate DDM-VTG’s ability to estimate uncertainty for unreasonable text queries through both qualitative and quantitative experiments:
> >
> > - **Qualitative Analysis:**
> >
> >     Figure 13 presents a representative case study involving mismatched text queries (lines 1162–1176).
> >
> > - **Quantitative Analysis:**
> >
> >     Figure 6 simulates unreasonable text queries by replacing a proportion of tokens in the original text with random tokens from other text queries in the batch. This provides a controlled evaluation of DDM-VTG’s uncertainty estimation for unreasonable or incoherent queries.

---

> > ### Author Response · Authors · 2024-11-23
> > **More Quantitative Results on Figure 2(a)**
> >
> > We provide supplementary quantitative results on the ActivityNet-CD and Charades-CD datasets to address the challenges outlined in Figure 2(a), as referenced in **Comment 5 to Reviewer QN6J**. Kindly feel free to review them.

---

> > ### Author Response · Authors · 2024-12-02
> > **Supplementary discussion (Addressing Weakness 3)**
> >
> > Dear Reviewer,
> >
> > As the discussion phase is nearing its end, we noticed that we have not yet received a response from you. To address any potential points of confusion and provide further clarification, we have revisited and reorganized our discussion on this issue.
> >
> > 1. **Addressing the Challenges in Figure 2:**
> > We have conducted a series of quantitative experiments involving VTG downstream tasks (Tables 1, 2, and additional results, such as Answer 5 to **Reviewer yg5S** comparing performance on large-scale datasets). Our experimental findings show that the proposed method achieves competitive performance relative to recent state-of-the-art VTG methods. This suggests that the uncertainty measurement approach itself is meaningful. While the primary focus of this study is on measuring uncertainty in VTG inference, we are pleasantly surprised to observe that our method also leads to improvements in VTG task performance. We believe this enhancement stems from the fact that modeling uncertainty strengthens the model’s ability to extract important information from the video.
> > 2. **Evaluating Uncertainty Measurement Across Modalities:**
> > To further validate that our uncertainty measurement method can effectively capture uncertainty introduced by the text and image modalities, we conducted adversarial experiments. By introducing varying levels of noise into the text and image modalities, we observed the resulting changes in uncertainty during inference. This setup directly simulates the challenges described in Figure 2. The uncertainty distributions (shown in Figure 6) indicate that our method is highly sensitive to adversarial samples in both modalities, which helps address the modality imbalance issue present in the baseline model.
> > 3. **Addressing the Issue of Video Length Distribution:**
> > As noted, videos of varying lengths are common in real-world applications. Reference [1] discusses how current VTG datasets are affected by the uneven distribution of video lengths, leading to potential bias since models often focus on specific time intervals where most localization occurs. To address this, we visualized the uncertainty inference results on QVHighlight (Figure 8) and conducted ablation studies. The results demonstrate that our method can effectively capture this issue through uncertainty measurement, with the model exhibiting significantly higher uncertainty for less frequent localization results. This further highlights that DDM-VTG exhibits strong capabilities in handling out-of-distribution (OOD) samples and mitigating location bias.
> >
> >     In response to the concerns raised by **Reviewer QN6J** and **Reviewer yg5S** regarding the generalization of DDM-VTG, we have added experiments on ActivityNet-CD and Charades-CD as suggested (please refer to Answer 5 to Reviewer QN6J). In these datasets, the IID samples represent data processed to address location bias, while the OOD samples are unprocessed. The results show that our method outperforms the baseline in OOD inference and achieves a smaller **IID-OOD Gap**, indicating that DDM-VTG is robust to location bias.
> >
> > 4. **Qualitative Case Studies:**
> > While we have presented numerous quantitative results, our primary goal is to demonstrate that the model can make robust inferences even when confronted with clearly anomalous inputs, as outlined in Figure 1. To visually demonstrate this ability, we have included a series of case studies (**Figure 11 in Appendix C.3**, and **Figures 13–16 in Appendix C.5**). These cases cover the challenges mentioned in Figure 2(a), (b), and (d). Since existing VTG methods do not explicitly quantify uncertainty to address these challenges, we chose UniVTG as a representative baseline for comparison. The case-by-case comparisons and analyses have been provided in the **rebuttal.zip**. These examples confirm that our model is able to sensibly and accurately express the uncertainty involved in inferences when facing these challenges.
> >
> >     However, due to the inherent limitations of the model's knowledge capacity, we observed that the model's inference on extreme samples still lacks fine-grained characterization. For example, similar extreme OOD samples tend to consistently show extreme uncertainty values, such as 0.99 or 1.00. We sincerely hope to further improve this aspect in future work, allowing the model to produce more nuanced and reliable uncertainty estimates, which would lead to more trustworthy VTG inferences.
> >
> >
> > [1] Yuan, Yitian, et al. A closer look at temporal sentence grounding in videos: Dataset and metric. *Proceedings of the 2nd international workshop on human-centric multimedia analysis*. 2021.

---

> ### Author Response · Authors · 2024-11-20
> **5. On the Definition of Prediction Accuracy in the Geom-Regularizer (Addressing Question 2)**
>
> We define prediction accuracy using a simple error metric, as detailed in **line 184**. In the context of the geom-regularizer, where accuracy must lie between 0 and 1, we apply a straightforward normalization technique. This is described in **lines 301–310** and further elaborated in **Appendix B.2**.

---

> ### Author Response · Authors · 2024-11-24
>
> Dear reviewer,
>
> We wonder if our response answers your questions and addresses your concerns? If yes, would you kindly consider raising the score? Thanks again for your very constructive and insightful feedback!

---

> ### Author Response · Authors · 2024-11-25
>
> Dear reviewer:
>
> With the discussion stage ending soon, we wanted to kindly follow up to check if our response has addressed your questions and concerns. If yes, would you kindly consider raising the score  before the discussion phase ends？ We are truly grateful for your time and effort！

---

> ### Author Response · Authors · 2024-12-02
>
> Dear reviewer,
>
> As the discussion stage, which has been extended by the ICLR committee, is ending soon, we wonder if our response answers your questions and addresses your concerns? If yes, would you kindly consider raising the score? Thanks again for your very constructive and insightful feedback!

---

> ### Author Response · Authors · 2024-12-03
> **Supplementary Discussion on Q1**
>
> We would like to thank you for your valuable feedback. In response to your concerns, we would like to clarify why our model, despite being trained on matched text-video pairs, is capable of measuring uncertainty when handling queries in an open-world setting, specifically for Video Temporal Grounding (VTG). The reasons for this capability are as follows:
>
> 1. **Introduction of Deep Evidential Regression (DER):** We leverage the Deep Evidential Regression (DER) technique, which learns a second-order distribution over the Gaussian parameters. This means that the model, when fitted to the training data, learns a higher-order distribution that allows it to perceive differences between new input samples (such as unmatched pairs) and the matched pairs seen during training. As a result, the model can express the reliability of its predictions through uncertainty measurements.
> 2. **Challenges in Utilizing DER for Explicit Uncertainty Measurement in VTG:** Despite the effectiveness of DER, directly applying it to VTG as a baseline for explicit uncertainty measurement presents two challenges:
>     - **Modality Imbalance:** There is a mismatch in the model's sensitivity to anomalous videos and queries. The model shows different levels of sensitivity to such outliers, which can lead to biased uncertainty measurements.
>     - **Bias in Uncertainty Estimation** We observed that the uncertainty estimated by the model during inference does not always adhere to the expected behavior: **"correct predictions should have low uncertainty, and incorrect predictions should have high uncertainty."** This suggests that the evidence and uncertainty learned by the model during training are misaligned, which in turn affects the higher-order distribution and leads to unreliable uncertainty estimates at inference time.
>
> To address these issues, we adopted a more refined modal alignment strategy and made structural improvements to the regularizer used in the vanilla DER (introducing a geom-regularizer). These changes ensure that the model accurately recognizes matched samples and can effectively distinguish unmatched samples with substantial differences.
>
> Through extensive experiments, we validate our uncertainty measurement method. On one hand, we found that this approach unexpectedly improves the model's performance on downstream tasks, likely due to its enhanced ability to perceive uncertainty. On the other hand, the model demonstrates an explicit ability to measure location bias and handle anomalous inputs in the open-world setting, thus achieving the intended design goal. Regarding the details above, we have further elaborated on them in the supplementary discussion titled **“Supplementary Discussion (Addressing Weakness 3)”**, which was provided yesterday.
> We would like to invite you to review it and let us know if you have any further questions. If we have successfully addressed your concerns, we would greatly appreciate your positive response at your convenience.

---

> ### Author Response · Authors · 2024-12-03
>
> Dear reviewer MSwQ:
>
> With the discussion stage ending soon, we want to kindly follow up to check if our response has addressed your questions and concerns. If yes, would you kindly consider raising the score before the discussion phase ends？ We are truly grateful for your time and effort！

---

### Author Response · Authors · 2024-11-24
**Summary of Revisions and Responses to Reviewer Feedback**

We sincerely thank all reviewers for their time and thoughtful feedback, as well as for recognizing the innovation in our work. The constructive comments have been invaluable in improving our paper.
In this rebuttal, we have addressed the reviewers’ concerns by conducting additional experiments, including comparisons with SOTA methods, ablation studies, and parameter analysis. We have also performed qualitative case studies to provide deeper insights.
Beyond the official comments, we have revised Figure 3 based on Reviewer FPPU’s suggestions, corrected minor typographical errors (marked in blue in the revised PDF), and uploaded the updated version. Additionally, following Reviewer FPPU’s recommendation, we have included more detailed case studies in the supplementary file （**rebuttal.zip**).
We welcome any further discussion and look forward to refining the paper further based on your feedback. Thank you again for your constructive and insightful reviews.

---

### Meta-Review · Area_Chair_u3dz · 2024-12-20

**Metareview:**

This work introduces a new model, DDM-VTG, that integrates Deep Evidential Regression into Video Temporal Grounding in order to handle uncertainties in open-world scenarios. The paper got diverse recommendations (two acceptance and two rejection). Though there are merits in this paper, the work has the following weaknesses which are raised by reviewers: (1)  lack of substantial theoretical and experimental analysis (2) performance improvement is not obvious (3) there are concerns about the used open world datasets. AC appreciates the contributions of this work, but still think the current version is not ready for publication at this top conference.

**Additional Comments On Reviewer Discussion:**

The authors provided additional experiments and analysis, and provided the motivations. However, the replies have not fully addressed reviewers' concerns.

---

### Decision · Program_Chairs · 2025-01-22

Reject